# Lead-free Zr-doped ceria ceramics with low permittivity displaying giant electrostriction

Maxim Varenik [1,5], Boyuan Xu [2,5], Junying Li[3], Elad Gaver [1], Ellen Wachtel[1], David Ehre [1], Prahlad K. Routh [3], Sergey Khodorov[1], Anatoly I. Frenkel [3] ✉, Yue Qi [4] ✉ & Igor Lubomirsky [1] ✉

Electrostrictors, materials developing mechanical strain proportional to the square of the applied electric field, present many advantages for mechanical actuation as they convert electrical energy into mechanical, but not vice versa. Both high relative permittivity and reliance on Pb as the key component in commercial electrostrictors pose serious practical and health problems. Here we describe a low relative permittivity (<250) ceramic, $Zr_xCe_{1-x}O_2$ (x < 0.2), that displays electromechanical properties rivaling those of the best performing electrostrictors: longitudinal electrostriction strain coefficient ~$10^{-16}$ m²/V²; relaxation frequency ≈ a few kHz; and strain ≥0.02%. Combining X-ray absorption spectroscopy, atomic-level modeling and electromechanical measurements, here we show that electrostriction in $Zr_xCe_{1-x}O_2$ is enabled by elastic dipoles produced by anharmonic motion of the smaller isovalent dopant (Zr). Unlike the elastic dipoles in aliovalent doped ceria, which are present even in the absence of an applied elastic or electric field, the elastic dipoles in $Zr_xCe_{1-x}O_2$ are formed only under applied anisotropic field. The local descriptors of electrostrictive strain, namely, the cation size mismatch and dynamic anharmonicity, are sufficiently versatile to guide future searches in other polycrystalline solids.

Materials developing mechanical strain of tens to hundreds parts per million (ppm), under applied electric field, are a backbone of essential technologies including actuators, sensors, and transducers[1,2]. The search for novel electromechanically active materials is ongoing in various directions, including domain engineering, induction of structural instabilities, and preparation of composites. For a number of transducer applications[3–5], electrostrictors, in which strain is linearly proportional to the square of the applied electric field, have an advantage over piezoelectrics, in which strain is linearly proportional to the field, as the former do not develop polarization under stress allowing for higher actuation accuracy and simpler driving electrical circuitry. However, since their first appearance during the 1980s, the

best electrostrictors are still based on $PbMg_{1/3}Nb_{2/3}O_3$ (PMN), often in solid solution with small amounts (<15 mol%) of $PbTiO_3$ (PMN-PT)[6,7]. The electrostriction strain coefficient of PMN-PT reaches $10^{-16}$ m²/V²[6,8] but its applicability is limited by lead-related toxicity, very high relative permittivity ($\varepsilon_{PMN} > 10000$ [8]), and poor compatibility with silicon microfabrication.

For aliovalent doped ceria ($CeO_2$), an electrostriction strain coefficient as large as that of PMN-PT ($10^{-16}$ m²/V²)[9,10], coexists with a much lower relative dielectric permittivity than that of PMN-PT as well as a much higher elastic modulus. However, only at very low frequencies (<1 Hz), where the relative dielectric permittivity is $30 < \varepsilon' < 500$, is the longitudinal electrostriction strain coefficient of

[1]Department of Molecular Chemistry and Materials Science, Weizmann Institute of Science, Rehovot 761001, Israel. [2]Department of Physics, Brown University, Providence, RI 02912, USA. [3]Department of Materials Science and Chemical Engineering, Stony Brook University, Stony Brook, NY 11794, USA. [4]School of Engineering, Brown University, Providence, RI 02912, USA. [5]These authors contributed equally: Maxim Varenik, Boyuan Xu. ✉e-mail: anatoly.frenkel@stonybrook.edu; yueqi@brown.edu; igor.lubomirsky@weizmann.ac.il

doped ceria comparable to that of PMN-PT. At higher frequencies, both the electrostriction strain coefficient and the dielectric permittivity decrease to $10^{-17} - 10^{-18}$ m$^2$/V$^2$ and <50, respectively. In both frequency ranges, this combination of low permittivity and high elastic modulus (i.e., weak compliance) places the hydrostatic electrostriction polarization coefficient $Q_h$ calculated for ceria at least two orders of magnitude above that predicted by Newnham's scaling law[11,12] and identifies it, as well as other recently described ceramics with a large concentration of point defects[12–17], as "giant", or non-classical electrostrictors[11,18]. These ceramics joined other groups of materials, such as hybrid perovskites[19,20] and polymer composites[21,22], which have been classified[18] as "giant" electrostrictors" by virtue of displaying an electrostriction coefficient that is at least one order of magnitude larger than that predicted by Newnham's scaling law.

In contrast to PMN-PT, doped ceria is non-toxic and fully compatible with Si-based microfabrication processes[13,23], has a high elastic modulus, as well as low dielectric permittivity. All three factors favorably distinguish ceria-based electrostrictors from other electrostrictive materials currently in use. However, its high longitudinal electrostriction strain coefficient decays rapidly above ≈1 Hz and the electric field-induced strain saturates at <15 ppm[10,24,25], while in PMN-PT, strain of hundreds of ppm can be achieved and its usable frequency range reaches a few kHz.

Examining the electrostrictive properties of trivalent lanthanide-doped ceria revealed that for 10 mol% dopant, the longitudinal electrostrictive strain presents a distinctly different dependence on dopant size at low (≤1 Hz) and at higher (≥100 Hz) frequencies[24], (Fig. 1a, b). The longitudinal, low frequency electrostriction strain coefficient ($M_{33}^{1\,Hz}$) measured for trivalent-doped ceria has been attributed to field-induced reorientation of elastic dipoles induced by oxygen vacancies, correlating with the well characterized intermediate temperature oxygen ion conductivity[26]. Elastic dipoles are defined as an anisotropic elastic field, capable of reorientation under external anisotropic stress[27]. In the case of aliovalent-doped ceria, elastic dipoles are present even in the absence of an external electric field. Once applied, the field is able to reorient these pre-existing, "static" elastic dipoles. Electrostrictive strain at higher frequencies (i.e., $M_{33}^{100\,Hz}$) increases exponentially with decreasing dopant radius[24] (Fig. 1b), suggesting an additional mechanism for electrostriction, one which is independent of dopant valence and consequently does not require the presence of oxygen vacancies. Here we report that $|M_{33}| \approx 10^{-16}$ m$^2$/V$^2$ for 10 mol% isovalent Zr$^{4+}$-doped ceria ceramics throughout the 0.15–3000 Hz frequency range, achieving strain >200 ppm without apparent strain saturation. In contrast to aliovalent doped ceria, X-ray absorption spectroscopy (XAS) and theoretical DFT modeling find no local deviation from cubic symmetry in the vicinity of the Zr ions within the host ceria lattice. This eliminates the possibility of pre-existing "static" elastic dipoles associated with the Zr dopant. Rather, due to bond anharmonicity, the local elastic field becomes anisotropic only upon application of an external field. Zr-O bonds were found to be shorter by ~0.1 Å than Ce-O bonds and highly anharmonic, due to the expanded range of motion available for [ZrO$_8$] local bonding units compared to the [CeO$_8$] host. These conditions give rise to "dynamic" elastic dipoles, i.e., elastic dipoles that are formed only under an external field due to anharmonicity, revealing a previously unknown mechanism of non-classical electrostriction.

## Results

### Crystal structure and elastic moduli

X-ray diffraction (XRD) profiles for Zr$_x$Ce$_{1-x}$O$_2$ (0 < x ≤ 0.2) ceramic pellets (10 mm diameter, 0.8-2 mm thickness), prepared as described in Methods, reveal randomly oriented crystallites with fluorite (Fm-3m) structure (Supplementary Fig. 1). The lattice parameter decreases with increase in Zr content due primarily to the smaller size of Zr$^{4+}$ relative to Ce$^{4+}$. The mean grain size (Supplementary Fig. 2[28]) decreases with Zr addition from 3 μm at 5 mol% to 1 μm at 10 mol%. Ceramic porosity was ≤ 6 vol%, required for accurate determination of Young's ($Y$) and shear ($G$) moduli via ultra-sound time of flight measurements (USTOF, Supplementary Note 3[29,30]).

### Electrostrictive strain measurements

Following our earlier work[24], we began this study with Zr$_{0.1}$Ce$_{0.9}$O$_2$. Ceramic pellets, re-oxidized to ≤100 ± 10% ppm Ce$^{3+}$ (see Methods for description of SQUID magnetometry; and Supplementary Note 4), displayed a direct longitudinal electrostriction strain coefficient ($M_{33} = u_{33}/E_3^2$ of · $10^{-16}$ ± 10% m$^2$/V$^2$). All samples described in this work contract along the field ($M_{33} < 0$), irrespective of the field direction (i.e., parallel or anti-parallel) (Supplementary Fig. 19), similar to previously reported aliovalent doped ceria[24,31], delta-phase Bi$_2$O$_3$[12], nominally dry or hydrated acceptor-doped BaZrO$_3$[17], and fluoride minerals[11]. In fact, none are auxetic, meaning all display positive Poisson's ratio. The dependence of strain on electric field, $u_{33}\ vs\ E_3^2$ remains near-linear for $E ≤ 13.4$ kV/cm, where $u_{33} ≈ −200$ ppm (Fig. 2a). The longitudinal electrostrictive strain remains linear with the applied electric field squared within the accessible range of fields (0–13.4 kV/cm). This indicates that the strain achieved (225 ppm) is far from saturation and that the material remains within the linear dielectric regime (i.e., polarization remains linearly proportional to the electric field). No frequency dependence was observed between 0.1–150 Hz for direct electrostriction measurements. Converse electrostriction measurement of the strain coefficient ($M_{33}, = \varepsilon_0 \cdot \frac{\varepsilon_{S=0} - \varepsilon_S}{S}$, see Methods and Supplementary Note 6) agreed with the direct measurements for $f = 10$–150 Hz (Fig. 2b) while revealing that $M_{33}$ undergoes 50% relaxation only at $f = 6$ kHz ($S$ is the uniaxial stress applied; $\varepsilon_{S=0}$ and $\varepsilon_S$ are the dielectric constants without and with the stress). The change in dielectric permittivity remained linear within the measured range of applied compressive stress (Figs. S6−2). A dielectric loss peak is also observed within this range of frequencies (Figs. S5−3). These results place 10 mol% Zr-doped ceria on a par with commercial 85/15 PMN-PT electrostrictors (TRS technologies[8]) with respect to the

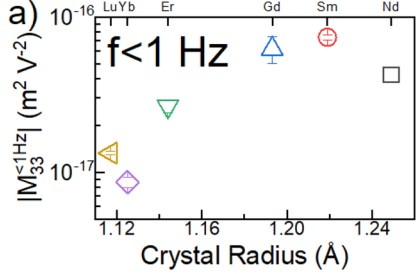

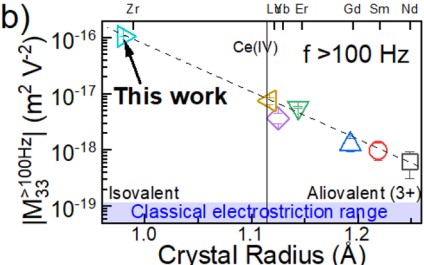

**Fig. 1 | Longitudinal electrostriction strain coefficient of 10 mol% doped ceria ceramics as a function of dopant crystal radius. a** $f < 1$ Hz and **b** $f > 100$ Hz. The data point for Zr in Fig. 1b extends the linear trend observed for trivalent-doped ceria. $M_{33} < 0$ (all samples contract parallel to the applied field). Data points for trivalent dopants are from ref. 10.

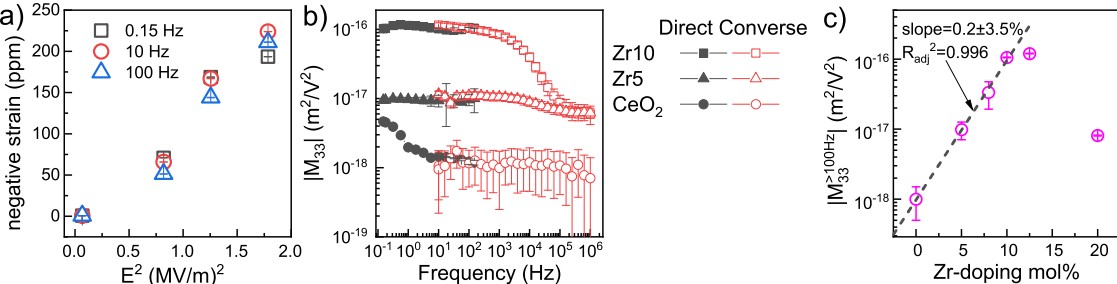

**Fig. 2 | Direct and converse longitudinal electrostrictive response in oxidized Zr-doped ceria ceramics. a** Longitudinal electrostrictive strain measured for $Zr_{0.1}Ce_{0.9}O_2$ as a function of the applied electric field squared. Values of $R^2_{adj}$ for the linear fit for strain measurements made at 0.15, 10 and 100 Hz are 0.97, 0.93 and 0.95, respectively. Typical amplitude for the AC voltage applied on the samples were between 100 and 1750 $V_{AC}$. **b** Log-log plot of the absolute value of the converse ($V_{AC} = 10$ V) and direct longitudinal electrostriction strain coefficients for undoped ceria and for 5–10 mol% Zr-doped ceria as a function of frequency. **c** Direct longitudinal electrostriction strain coefficient of Zr-doped ceria as a function of Zr content, $f \geq 100$ Hz. Measurements were made in triplicate for each sample (>4 samples for each composition) under ambient conditions; in some cases, error bars are smaller than the symbols. All samples contract parallel to the applied field.

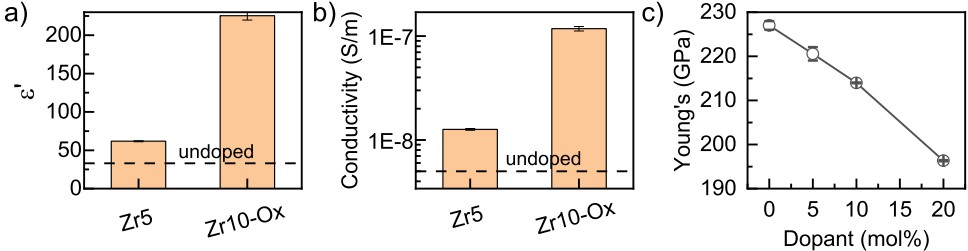

**Fig. 3 | Electrical and mechanical measurements of oxidized Zr-doped ceria ceramics. a** Real component of the relative dielectric permittivity and **b** total conductivity of oxidized ceria ceramics containing 5–10 mol% Zr as determined using impedance spectroscopy under ambient conditions; for IS spectra, see Supplementary Note 4. Measurements were made at $V_{AC} = 10$ volts, $f = 100$ Hz, with stainless steel electrodes, the upper one being spring loaded. Two pellets with the same chemistry were each measured in triplicate. **c** Young's modulus determined by USTOF measurements (see Methods section for details) as a function of Zr content 0-20 mol%. In some cases, the error bars are smaller than the data symbol.

electrostriction strain coefficient, saturation strain and relaxation frequency. Aiming to maximize $M_{33}$, we investigated its dependence on x: *i.e.*, x = 0.05, 0.075, 0.1, 0.125 and 0.2. For x ≤ 0.1 (Fig. 2c). The longitudinal electrostriction strain coefficient increases exponentially with Zr-content in the 0–10 mol% range. A plateau is observed between x = 0.1–0.125; followed by a sharp decrease for x = 0.2. However, the re-oxidation protocol becomes increasingly ineffective for x > 0.1 and the concentration of $Ce^{3+}$ for x = 0.125 or 0.2 is considerably larger than 100 ppm measured for x ≤ 0.1.

## Dielectric and elastic properties

Impedance spectra of the re-oxidized ceramic pellets, measured under ambient conditions as described in Methods, displayed only part of a circular arc on a Nyquist plot (Supplementary Note 5), from which both components of the complex permittivity were estimated. At 100 Hz, in the absence of dopant, $\varepsilon^{100Hz}_{CeO_2} \approx 32 \pm (5\%)$, a value similar to that of pure $ZrO_2$. However, in contrast to a rule of mixtures predictions[32], doping with 5-10 mol% Zr produces a marked increase in both the real and imaginary components of the relative permittivity, $\varepsilon^{100Hz}_{Zr_{0.05}Ce_{0.95}O_2} = 62 \pm 1$ and $\varepsilon^{100Hz}_{Zr_{0.1}Ce_{0.9}O_2} = 224 \pm 2$ (Fig. 3a). Conductivity at 100 Hz (Fig. 3b) increases even more dramatically with respect to $CeO_2$, up to two orders of magnitude for $Zr_{0.1}Ce_{0.9}O_2$. We have also measured the Young's modulus[33], $Y$, using USTOF as described in Methods, and have observed decrease of ≈5.7% for $Zr_{0.1}Ce_{0.9}O_2$ (Fig. 3c) and ≈13% for $Zr_{0.2}Ce_{0.8}O_2$ (Fig. 3c). This decay is unexpected since Zr-doping does not introduce oxygen vacancies and, therefore, does not reduce the number of chemical bonds in the lattice. The Young's modulus of $Zr_{0.1}Ce_{0.9}O_2$ is the same as that of $Gd_{0.1}Ce_{0.9}O_{1.95}$, even though 2.5% of the oxygen sites are unoccupied in the latter[34]. Increase in the relative permittivity (and in turn measured conductivity) is possible due to two mechanisms: (i) decrease in the elastic modulus suggest significant chemical bond weakening, which will increase permittivity as predicted by Kramers–Kronig relations; (ii) increase in the concentration of $Ce^{3+}$, which will increase the concentration of electrons in the conduction band. While the increase in the relative permittivity is consistent with the appearance of a localized Zr-4d electronic state within the $CeO_2$ band gap, predominantly between O-2p and Ce-5d orbitals, the decrease in elastic modulus cannot be correlated with any electronic structure change[35] (Figs. S8–3). These findings prompted experimental investigation of the local environment of Zr using X-ray absorption spectroscopy (XAS), supported by density functional theory (DFT) - based modelling and ab initio molecular dynamics (AIMD) calculations. XAS was shown to be successful in identifying point defect-induced elastic dipoles responsible for electrostriction in aliovalent-doped ceria[17,36–42].

## Local environment of Zr in $Zr_xCe_{1−x}O_2$: XAS and DFT modelling of the steady state

The average XRD structure of $Zr_xCe_{1−x}O_2$ ceramics is single-phase fluorite, where each cation is bound to 8 anions. Ce $L_3$-edge X-ray absorption near edge structure (XANES) spectra measured under ambient conditions for Zr-doped ceria, x = 0.05, 0.10 or 0.20, point to the same (fluorite) local structure of Ce environment, as revealed by the characteristic double white line of $Ce^{4+}$ (Fig. 4a)[40,43]. The putative presence of $Ce^{3+}$ at the rising edge of the Ce XANES spectrum could not be detected for any of the samples. The local environment of Zr atoms is analyzed by combining the Zr K-edge XANES and the extended X-ray absorption fine structure (EXAFS). The EXAFS spectra in k-space and Fourier-transformed to r-space are presented in Fig. 4b, c, respectively.

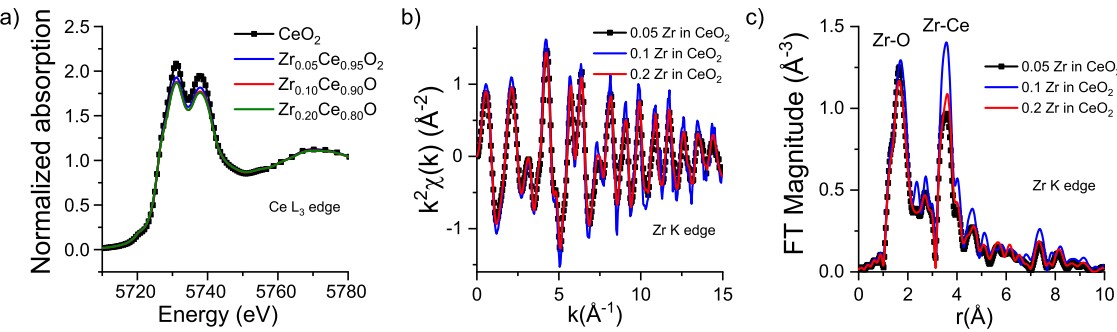

**Fig. 4 | X-ray absorption data for oxidized Zr-doped ceria ceramics.** Normalized Ce L₃ edge XANES spectra (**a**) and Zr K-edge EXAFS spectra in k-space (**b**) and r-space (**c**) for 5, 10 and 20 mol% Zr-doped ceria powders. r-space spectra were obtained by Fourier transforming the $k^2$-weighted $\chi(k)$ spectra in the $k$ range 3–14.5 Å⁻¹. The XANES spectrum of undoped ceria powder is included as a reference in (**a**). Details of XAS measurements are presented in the Methods section.

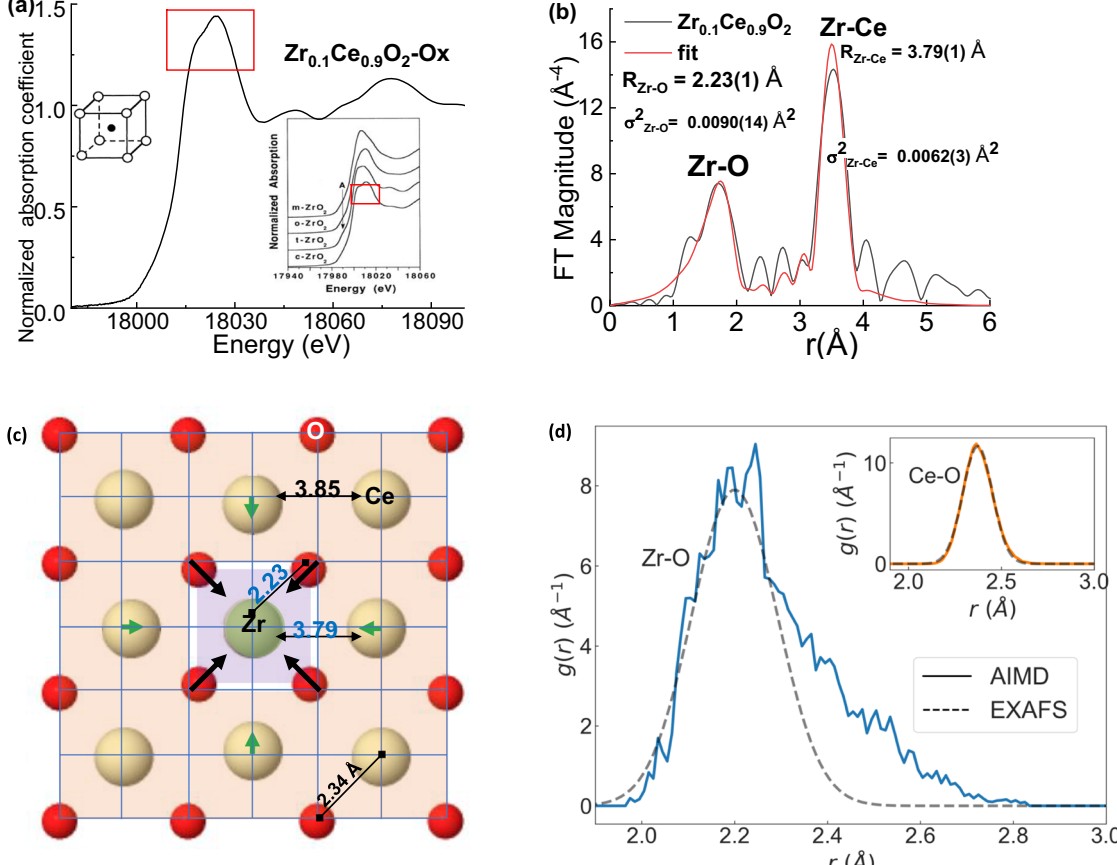

**Fig. 5 | Cation – oxygen bond distances in Zr-doped-CeO₂. a** Normalized Zr K-edge XANES spectrum of 10 mol% Zr in oxidized ceria powders and (inset) four polymorphs of Zr oxide[54]. Red rectangles are added as a guide to the eye. **b** Magnitude of the Fourier transform of $k^3$-weighted EXAFS spectra plotted as a function of r(Å) (black line); theoretical fit using the Zr-O and Zr-Ce photoelectron paths (red line). $R_{Zr-O}$ and $R_{Zr-Ce}$ refer to the bond length of Zr-O and inter-cation distance Zr-Ce, respectively. **c** Local structure around Zr from EXAFS analysis. Cations are in yellow; oxygens are in red. **d** ab initio molecular dynamics (AIMD) simulated radial distribution function (solid lines) for Zr-O of 1 Zr in 2 × 2 × 2 CeO₂ system and (inset): Ce-O in CeO₂. The dashed lines are Gaussian approximations for harmonic Zr-O and Ce-O bond length distributions as determined by EXAFS. Details of AIMD calculations are presented in the Methods section.

The first and second peaks (corresponding to Zr-O and Zr-Ce, respectively) are well-isolated from other contributions for all values of x (Fig. 4c). Their positions (after correcting for the photoelectron phase shift that causes the peaks to appear at distances ca. 0.3–0.5 Å lower than in real space) are consistent with Zr substituting for Ce in the lattice. This conclusion is further confirmed by the white line of the Zr K-edge XANES spectrum of Zr₀.₁Ce₀.₉O₂ that displays a weakly resolvable double-peak (red rectangle in Fig. 5a, and Supplementary Fig. 13) consistent with the presence of quasi-cubic [ZrO₈] structural units, and by the results of Zr K-edge EXAFS analysis that are described below.

Nonlinear least squares fitting of Zr K-edge EXAFS spectra (Fig. 5a, b), using the Zr-O and Zr-Ce photoelectron single-scattering paths in the model, gives Zr-Ce distance of 3.79 ± 0.01 Å which is 0.06 Å shorter than the Ce-Ce distance (3.85 Å). The Zr-O bond length (2.23 ± 0.01 Å) is shorter than the Ce-O bond length (2.34 Å)[36] by 0.11 Å. The difference of 0.11 Å - 0.06 Å = 0.05 Å implies that [ZrO₈] units have extended range for displacement within the lattice compared to the

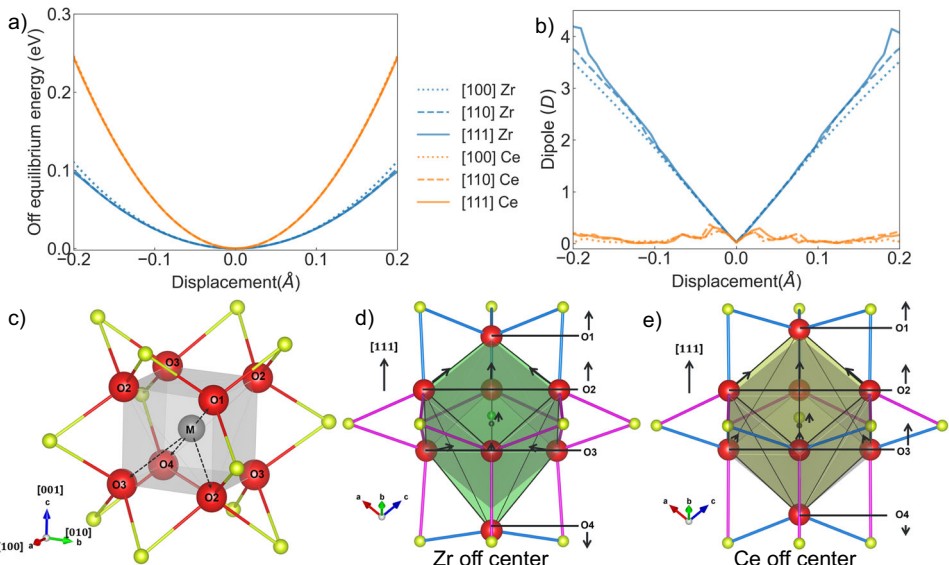

**Fig. 6 | DFT computed distortion of a cation-$O_8$ cube in Zr-doped CeO$_2$. a** The off-equilibrium energy difference for a cation-$O_8$ cube, MO$_8$ (M=Zr or Ce) displaced along different crystallographic directions. **b** The supercell electric dipole induced by off-centered MO$_8$. The electric dipole unit is debye (D), where $1D \approx 0.2082e \cdot$ Å. **c.** Schematic illustration of the MO$_8$Ce$_{12}$ local structure of M-O bonds fall into 4 categories based on their distortions (O1 to O4, red color). Cations are in yellow; oxygens are in red. The off-equilibrium MO$_8$Ce$_{12}$ structures viewed parallel to the [$\bar{2}$11] direction with **d** Zr as the center green atom and **e** Ce as the center yellow atom, each cation displaced by 0.22Å parallel to the [111] direction. Arrows represent the direction of oxygen displacements, and the blue (pink) bond color signifies

O-Ce bond length contraction (elongation). As M moves up, O1 and O2 atoms follow M by also moving parallel to the [111] direction. O4 moves away from M parallel to the [$\bar{1}\bar{1}\bar{1}$] direction (the O4 displacement for Zr is 20 times larger than that for Ce), for both Zr and Ce. The major difference between the center cations is in the behavior of the O3 atoms. Instead of following the Ce atoms, the O3 atoms displayed a displacement component that increased the distance from the Zr center atom. As a result, the Ce-O3 bond lengths were shortened parallel to the [111] direction in the Ce-centered structure while the Ce-O3 bonds were elongated in the Zr-centered structure. (Supplementary Table 4).

[CeO$_8$] host. Consistent with the EXAFS results, the DFT relaxed structure (0 K) revealed that the Zr-O bond length (2.28 Å) is 0.11 Å shorter than the next nearest neighbor (NNN) Ce-O bond length (2.39 Å). This result is expected for an isovalent substituent ion of smaller radius than the host.

The elastic dipole tensor for Zr as dopant, containing both local and long range deformation, was also computed using DFT[26,44]. Combining the elastic dipole tensor with the elastic stiffness tensor (Supplementary Note 8.2), we obtain the dopant - induced strain tensor per Zr-ion, $\boldsymbol{\alpha}_C$, which is isotropic with all diagonal terms $\alpha_{C,0K,ii} = -0.0657$. The negative values indicate a macroscopic volume shrinkage in CeO$_2$ due to Zr doping and the overall strain, $\boldsymbol{u}_C$, is a linear function of Zr-doping concentration, $C$, as $\boldsymbol{u}_C = \boldsymbol{\alpha}_C C$. This is close to the experimental values (Supplementary Fig. 1), demonstrating the adequacy of the model. Neither EXAFS nor DFT find any significant bi-modal dopant-O or host-O bond lengths. Thus, in contrast to the case of aliovalent doped ceria[37], the local symmetry in $Zr_xCe_{1-x}O_2$ (x < 0.1) is not reduced; there are no static elastic dipoles, such as those observed and successfully modeled in oxygen deficient ceria[26].

DFT-based AIMD simulations at 300 K were performed for CeO$_2$ with and without Zr (SI Supplementary Fig. 16). The average displacement of Zr away from its equilibrium site is 0.14 Å, i.e., 27% larger than for Ce. The radial distribution function (Fig. 5d) characterizes Zr-O bonds as strongly anharmonic, while Ce-O is harmonic. The observation of strong asymmetry in the Zr-O bond distribution is a key for reconciling the difference between the apparent, low variance of the Debye Waller factor $\sigma^2$ in the Zr-O distance measured by EXAFS (0.009 Å$^2$, Supplementary Table 3) and the large Zr-O $\sigma^2$ calculated by DFT (0.024 Å$^2$, Supplementary Fig. 16b). Numerous studies have demonstrated that EXAFS fitting analysis that assumed harmonic bonds would underestimate $\sigma^2$ for anharmonic bonds by a factor of up to 3-4[39,45-47]. In this study, the values of the Zr-O $\sigma^2_{EXAFS}$ obtained by

EXAFS are smaller by a factor of 2.7 than the Zr-O $\sigma^2_{AIMD}$ values, consistent with the large anharmonicity of Zr-O bonds. Harmonic Ce-O bonds (Fig. 6d inset), on the contrary, have similar values for $\sigma^2_{EXAFS}$ and $\sigma^2_{AIMD}$ (0.008 ± 0.001 Å$^2$ and 0.006 Å$^2$, Supplementary Fig. 16b, respectively). It is therefore evident that AIMD results uncover a much more fluxional [ZrO$_8$] environment compared to the more rigid one predicted by EXAFS based on the inability of the latter method to adequately probe strongly asymmetric distributions. We note one commonality with the previously described NCES material, Gd-doped ceria, in which the bond length distortion of the [CeO$_7V_O$] units was also characterized by significant anharmonicity[39]. This, however, is not valid for $Zr_xCe_{1-x}O_2$ (x < 0.1) because vacancy concentration is negligible, demanding an alternative NCES mechanism in Zr-doped ceria.

**Dynamic elastic dipoles in $Zr_xCe_{1-x}O_2$**

Snapshots made during DFT-based AIMD simulations at 300 K reveal a non-isotropic Zr dopant induced strain tensor, (Supplementary Table 6), suggesting the presence of ***dynamic*** elastic dipoles. The large displacement of Zr from its fluorite lattice site results in distortion of both [MO$_8$] cubes, (M = Zr or Ce). The $ZrO_8$ cube bond and angle distortion parameters are more distorted than [CeO$_8$] distant from Zr. The loosely fitting Zr and its large amplitude anharmonic motion suggested that it can be readily displaced by an external electric field. Indeed, we computed the energy landscape for the off-centered [MO$_8$], by displacing the M-atom parallel to the [100], [110], or [111] directions (Fig. 6a) allowing all oxygen atoms to relax while fixing cation positions. The energy cost to shift [ZrO$_8$] within the large Ce second coordination cage is much lower than that of [CeO$_8$](Fig. 6a). The fitted stiffness constant for moving [ZrO$_8$] around its equilibrium site is only slightly (20%) anisotropic. Stiffness for moving [CeO$_8$] in the bulk is essentially isotropic and 2 to 2.4 times higher than [ZrO$_8$]. Stiffness for moving the [CeO$_8$] complex first nearest neighbor to Zr, is only slightly decreased from that in the bulk. These provide the

theoretical basis for the more active Zr vibration and the reduction of the Young's modulus with Zr doping.

The dynamic elastic dipole and strain induced by a distorted $[MO_8]$ inside a larger Ce cage was further characterized by moving the cation ≈ 0.22 Å (the maximum allowed displacement at 300 K) parallel to the most labile ([111]) direction (i.e., parallel to the Zr/Ce-O bonds, Fig. 6d, e; also, Supplementary Fig. 18, Supplementary Table 4). The off-centered $[ZrO_8]$ led to a much larger computed electric dipole moment (in unit Debye) than the off-centered $[CeO_8]$ (Fig. 6b), tentatively explaining the increase of the relative permittivity resulting from Zr doping. The local distortion pattern of $[ZrO_8]$ (Fig. 6d) creates an anisotropic, defect-induced strain tensor (Supplementary Table 6). Comparing to the isotropic and time averaged $\alpha_{C,0K,ii} = -0.0657$, a net strain tensor can be diagonalized to show the principal strains of (0.145, -0.055, -0.054) per off-centered Zr. We propose that electric field-induced alignment of the dynamic elastic dipoles, is the source of the increased electrostriction coefficient. In addition, the dielectric relaxation observed in the frequency range of a few kHz is due to the inability of the induced elastic/electric dipole to follow the alternating electric field. In a sample of 10 mol% Zr-doped ceria with dynamic elastic dipoles, complete alignment of all the elastic dipoles parallel to the applied electric field or uniaxial mechanical stress may generate strain of ≈ 0.1·0.145 = 1.45% or 14500 ppm. This is ~60 times larger than the electrostrictive strain measured with the E-field <13.4 kV/cm used here. This is consistent with our estimate of an upper bound on Zr displacement. Since no strain saturation is observed in Fig. 2a, we suggest that not all elastic dipoles are aligned under the field.

## Discussion

Measurements of electrostrictive strain, presented above, demonstrate that Zr-doped ceria ceramics can rival PMN-PT, currently the best commercially available electrostrictor. The value of $|Q_h|$, the hydrostatic electrostriction polarization coefficient, predicted for a classical electrostrictor with dielectric permittivity and elastic modulus of $Zr_{0.1}Ce_{0.9}O_2$[11,12] is ≈ $0.07\, m^4/C^2$, which is more than two orders of magnitude lower than the actual measured value, 9.7 $m^4/C^2$, thereby identifying Zr-doped ceria as a non-classical electrostrictor (NCES).NCES was previously observed in intermediate temperature ionic conductors, e.g., aliovalent doped-$CeO_2$[10,24,40], (Y, Nb)-stabilized δ-phase (cubic) $Bi_2O_3$[12], acceptor doped $BaZrO_3$[17], and $La_2Mo_2O_9$[48]. For these ceramics, the origin of the NCES has been identified[26] as field-driven reorientation of elastic dipoles resulting from local symmetry breaking by point defects, i.e., oxygen vacancies for $CeO_2$, δ $- Bi_2O_3$ and nominally dry acceptor-doped $BaZrO_3$, or by proton interstitials in hydrated, acceptor-doped $BaZrO_3$. This explanation does not suit isovalent doping of ceria ceramics, e.g., $Zr_{0.1}Ce_{0.9}O_2$: an appreciable concentration of mobile point defects is lacking. An isovalent cation dopant, Zr, with crystal radius >8% smaller than the host cation, creates in the fluorite structure of $CeO_2$ a relatively small $[ZrO_8]$-bonding unit. The local relaxation of the host cations in the second coordination shell does not compensate for the smaller size of the $[ZrO_8]$-bonding unit, allowing the $[ZrO_8]$-unit with freedom to move with strong anharmonicity and to deform (Fig. 6d) with little energy cost (Fig. 6a), thereby forming a dynamic elastic dipole. Only when polarized by an external electric field does the dynamic elastic dipole produce anisotropic long range elastic strain. Since $[ZrO_8]$ units are easily polarizable (see Fig. 6b), the external electric field displaces them from the equilibrium position, generating a macroscopic mechanical strain. Because the elastic dipole is dynamic, the electrostriction strain relaxation frequency in Zr-doped ceria can reach the kHz range, unlike Gd-doped ceria, where strain relaxes at a few Hz because it is caused by the static elastic dipoles of the oxygen vacancies. The relaxation frequency is also somewhat lower than expected for the dynamic motion of a cation (such as in PMN-PT, Supplementary Fig. 21). This may perhaps be the result of the involvement of the oxygen coordination shell. Although

simple, this paraelectric, "dynamic" elastic dipole model can account for major experimental observations. Since dynamic elastic dipoles occupy only a small fraction of the crystal lattice (controlled by dopant concentration), they allow coexistence of a large electrostriction strain coefficient with a large elastic modulus and relatively low permittivity. In fact, this NCES mechanism may be more generally applicable. A small dopant cation, along with nearest neighbor anions, vibrating anharmonically in a relatively large (compared to the host cation) cage may facilitate electromechanical coupling. Since the square faces of $[MO_8]$ are less stiff than the triangular faces of an octahedron in perovskites or a tetrahedron in sphalerite, a fluorite lattice may be particularly suited to large dopant-host cation size mismatch. Therefore, we suggest that this NCES mechanism may not be limited to Zr-doped ceria, but may be realized in other fluorite-structured hosts as well, for instance, in Ca-, Sr- and Ba-fluorides.

## Methods

### Ceramic sample preparation

$Zr_xCe_{1-x}O_2$ ceramics in the shape of 10 mm diameter, 0.8-2 mm thick pellets were prepared via the rapid sintering protocol, as previously described[24,49]. This protocol prevents possible cation segregation due to limited mutual solubility, leading to a single fluorite phase. The porosity of the sintered pellets was deduced from the mass density as measured by the conventional Archimedes technique. All pellets were mirror polished and top and bottom faces were made parallel. Unless specified otherwise, the pellets were heated at 773 K for 5 h in pure oxygen to compensate for possible oxygen loss during sintering. Following re-oxidation, $Zr_xCe_{1-x}O_2$ (0.05 <x ≤ 0.2) pellets changed color from green-black, immediately following sintering, to yellow-white. The green-black coloration of the low concentration (x = 0.05) pellets was very weak.

### Determination of the $Ce^{3+}$ content from magnetization data

Zr-doping is known to promote reduction of $Ce^{4+}$ to $Ce^{3+}$, which is accompanied by the formation of oxygen vacancies. Since samples contained more than 99.9% $Ce^{4+}$, the concentration of $Ce^{3+}$ was determined from magnetization curves acquired with a Superconducting Quantum Interference Device (SQUID) Magnetometer (see SI section 7 for details); $Ce^{3+}$ is known to be magnetic, while $Ce^{4+}$ only accounts for weak temperature independent Van Vleck magnetization[50]. The concentration of $Ce^{3+}$ can be deduced from magnetic saturation (M) curves, after correcting for sample impurities, by fitting M(H) at a given temperature to a Langevin-type equation:

$$M = Ng\mu_B J \cdot L(\eta) + \chi_0 \cdot H \tag{1}$$

where N is the number of magnetic species per unit volume ($m^{-3}$); g is the Landé g-factor; $\mu_B$ is the Bohr magneton; $J = |L \pm S|$ and $L(\eta)$ is the Langevin function $L(\eta) = \coth(\eta) - \frac{1}{\eta}$, where $\eta$ is the ratio of the magnetic to thermal energy, $\eta = \frac{g\mu_0\mu_B J}{k_B T} \cdot H$; $\mu_O$ is the vacuum magnetic permeability; $k_B$ is the Boltzmann constant; T is absolute temperature (K); and H is the magnetic field strength (A $m^{-1}$), $\chi_0$ is a temperature independent contribution which accounts for diamagnetic and Van Vleck susceptibility. The value of $\chi_0$ was taken from previous studies[50].

### Measurements of the direct electrostriction effect: field-induced strain

Longitudinal (i.e., parallel to the applied electric field) electrostrictive strain, $u_{33}$, was measured with instrumentation described previously[10,34]. Briefly, the ceramic pellet was inserted between two stainless steel electrodes, with the top electrode being spring loaded. The displacement, δ, of the top surface under voltage was measured with a proximity sensor (±0.02 nm) monitored with a lock-in amplifier. Alternating sine-wave voltage was applied to the sample and only

second harmonic response was detected. The electrostriction coefficient in such a configuration is given by: $M_{33} = \delta \cdot th / V^2$, where $V$ is the voltage applied and $th$ is the sample thickness measured with accuracy $\pm 2\,\mu m$. The values of strain and electric field were calculated as $u = \frac{\delta}{th}$ and $E = V/th$. The measurements were performed under ambient conditions ($297 \pm 2$ K, relative humidity 20–55%). Typical amplitude for the AC voltage applied on the samples were between $100\,V_{AC}$ to $1750\,V_{AC}$, and typical sample thickness as 1 mm. All samples contract parallel to the applied field. Commercial samples of PMN-PT with silver contacts (TRS Technologies) and a 100-cut quartz single crystal without additional sputtered metal contacts were used for calibration of the measurement setup. Values matching literature data were obtained: $M_{33}(PMN - PT) = (3.5 \pm 0.5)10^{-16} m^2/V^2$ for PMN-PT and $d_{33}$(100 quartz) = $2.3 \pm 0.2$ pm/V within the frequency range 0.15–1000 Hz.

### Impedance spectroscopy and converse electrostriction

Impedance spectroscopy measurements were conducted at 298 K with a Novocontrol Alfa dielectric analyzer in the high voltage mode within the frequency range 1 mHz-1 MHz under excitation voltage of $U_{AC} = 10$ V (electric field ≈10 kV/m for all samples). Constant bias was not applied ($U_{DC} = 0$). Measured impedance values fall within the 1% accuracy range of the impedance analyzer. Conductivity, $\kappa$, and the relative permittivity, $\varepsilon_r$, were calculated as:

$$\kappa = \frac{Z_{Re}}{|Z|^2} \frac{th}{A_c} \text{ and } \varepsilon_r = -\frac{Z_{Im}}{2\pi f \cdot \varepsilon_0 \cdot |Z|^2} \frac{th}{A_c} \quad (2)$$

where $th$ is the pellet thickness, $A_c$ is the area of the contacts, $Z = Z_{Re} + iZ_{Im}$ is the complex impedance, and $f$ is the frequency and $\varepsilon_0$ is the vacuum permittivity. Converse electrostriction was measured with the impedance analyzer and a screw vise. The longitudinal electrostriction strain coefficient was determined from the change in the real component of the permittivity of the ceramic pellet under compressive stress, $s$:

$$M_{33} = \varepsilon_0 \cdot \frac{\varepsilon_{\sigma=0} - \varepsilon_\sigma}{s} \quad (3)$$

The value of the force applied with a screw vise, $F_{app}$ was monitored with a digital force sensor and the stress calculated as $s = F_{app}/A_c$ (for details, see SI section 6).

### X-ray absorption spectroscopy

XAS spectra at the Zr K-edge and Ce $L_3$-edge were collected in fluorescence mode from 50 micron (300 mesh) powders (made by grinding pellets that had been used in electrostriction measurements) at the QAS (7-BM) beamline at the National Synchrotron Light Source-II (NSLS-II) at Brookhaven National laboratory. Experimental details are available in Supplementary Note 7. Data analysis was carried out using Athena and Artemis programs, which are part of the Demeter data analysis package[51]. For each element (Zr or Ce) absorption edge, multiple spectra (up to 30 scans) were averaged, to increase the signal-to-noise ratio. The X-ray absorption near edge structure (XANES) data of Zr K-edge and Ce $L_3$-edge were pre-edge subtracted and edge-step normalized, and extended X-ray absorption fine structure (EXAFS) data were obtained following established procedures. FEFF6 theoretical code was used for calculating photoelectron scattering amplitude and phase shifts. The data were fit using a non-linear, least square Levenberg-Marquardt method in r-space, using Fourier transform of both data and theory. For non-linear fitting of the theoretical EXAFS equations to experimental spectra, corrections to the model Zr-O and Zr-Ce distances ($\Delta r$), variance, i.e., mean square relative displacements ($\sigma^2$) of $\Delta r$, and the correction to the photoelectron energy origin ($\Delta E$) – the same for the Zr-O and Zr-Ce paths - were varied in the k-range and r-range intervals of 3–14.5 Å$^{-1}$ and 1.5–3.9 Å, respectively. Third

cumulants of the radial distribution function were also varied but these did not affect the fit quality. EXAFS data of Zr foil were fit (Supplementary Fig. 14 and Supplementary Table 2) to obtain the amplitude reduction factor (0.99) which was then applied in fitting the spectra of the Zr-doped ceria (Supplementary Fig. 15 and Supplementary Table 3).

### Theoretical Modeling

DFT calculations were performed to connect the local lattice distortion with the long-range elastic strain induced by dilute Zr-dopants in $CeO_2$ via the elastic dipole tensor calculation. The point defect induced strain tensor per defect is defined as a coefficient, $\boldsymbol{\alpha_c}$. Due to a requirement for dilute formulations, the Zr-doping concentration was 3% (1 out of 32) in the DFT calculations. DFT calculations implemented in the Vienna Ab initio Simulation Package (VASP) were used. The generalized gradient approximation (GGA) of Perdew, Burke, and Ernzerhof (PBE)[52] was used for the DFT exchange correlation functional with the Hubbard-U correction ($U_{eff} = 4.5$ eV) for Ce 4f orbitals[53]. For AIMD calculations, the NVT ensemble was prepared at T = 300 K, and the original volume, maintained at the DFT- minimized value in order to compare the dynamic effect, was used. The structure characters were averaged for 4 ps with a time-step of 1 fs. The elastic dipole and the point defect induced strain tensor per defect, $\boldsymbol{\alpha_c}$ were computed following the method derived by Gillian[44] and the procedures provided by Das et al.[26]. (More details may be found in Supplementary Note 8).

## Data availability

The data that support the findings of this study are available from the corresponding author upon reasonable request.

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

## Acknowledgements

The work on development of novel electrostrictive materials was supported by the US Army Research Office (ARO grant #W911NF2110263, I.L.). The work on development of the theoretical description and the properties of the elastic dipoles was supported by the Israel-US Binational Science foundation regular program (Y.Q. + I.L grant # 2020108). IL and AIF acknowledge the NSF-BSF program grant 2022786, which supported the synchrotron measurements and their analysis. AIF and PKR acknowledge support by NSF Grant number DMR-2312690. This research used beamline 7-BM (QAS) of the National Synchrotron Light Source II (NSLS-II), a U.S. DOE Office of Science User Facility operated for the DOE Office of Science by Brookhaven National Laboratory under contract no. DE-SC0012704. We gratefully acknowledge Drs. Lu Ma and Steven Ehrlich for their support of the XAS measurements at the 7-BM beamline. We acknowledge support of the beamline experiments by the

Synchrotron Catalysis Consortium funded by the US Department of Energy, Office of Science, Office of Basic Energy Sciences, Grant No. DE-SC0012335. AIF acknowledges support by Weston Visiting Professorship during his stay at the Weizmann Institute of Science.

## Author contributions

A.I.F., Y.Q., I.L. conceived the study framework. M.V. performed the electrostriction, mechanical, magnetic, and electrical experiments and analysis on doped ceria samples with the help of E.G. S.K. performed the electrostriction experiments and analysis of PMN-PT. B.X. performed and analyzed the DFT and AIMD simulation. J.L. and P.K.R. performed and analyzed the x-ray absorption experiments. D.E. and E.W. contributed to experimental design and discussion. A.I.F., Y.Q., I.L. wrote the manuscript with contributions from all the authors.

## Competing interests

The authors declare no competing interests.
