## [Peer Review File · Nature Communications]

Lead-free Zr-doped ceria ceramics with low permittivity displaying giant electrostrictionEditorial Note: Parts of this Peer Review File have been redacted as indicated to remove third-party material where no permission to publish could be obtained.

REVIEWER COMMENTS

Reviewer #1 (Remarks to the Author):

So called “giant electrostrictors” have so far suffered from meagre responses at frequencies above 100Hz. As such their ability to be integrated in actuators is limited.

Electrostriction coefficient M of the order of $10^{-16} \text{ m}^2/\text{V}^2$ is at par with relaxor ferroelectrics but still below what has been reported e.g. for hybrid perovskites and composites. In addition, the developed strain is also of the same order of magnitude than PMN-PT.

Therefore, the use of “giant electrostriction” in the title seems not to be appropriate.

The advantage of Zr-doped CeO₂ compared to the composites is the stress developed thanks to their superior mechanical properties. Such advantage may be mentioned to underline the potential of this material in efficient actuators and as a credible alternative to PMN-PT.

The first main point to clarify is what an elastic dipole is for the authors. According to Kroner (Kröner, E. (1958). "Kontinuumstheorie der Versetzungen und Eigenspannungen," Springer, Berlin.) it is a point defect producing local distortions that react to a homogenous stress applied to the material in an analogous way an electric dipole reacts to an applied electric field (albeit one is a second-rank tensor and the other one a vector). As such, the strain tensor induced by the shorter Zr-O bonds (within the preserved fluorite structure) constitutes an elastic dipole. Hence, there should be what the authors call “equilibrium” elastic dipoles, even in the absence of an applied electric field.

Would the authors define what an elastic dipole is for them ?

Would the authors explain why they state there is no equilibrium elastic dipole ?

I recognise that the nature of the defect causing the elastic dipole is different from the one involving an oxygen vacancy but, to me, both lead to a local deformation susceptible to give

rise to an elastic dipole.

Fig2.c shows a linear relationship between the log of the high frequency M33 and the Zr doping content up to 10mol%. How do the authors relate this evolution to the number of elastic dipoles ? Why is it exponential and not linear ?

The question of the coupling mechanism to the electric field also remains to be clarified.

As an elastic dipole is defined by its reaction to a mechanical stress. Do the authors consider that the electric field induces a polarisation that, in turn, induces a stress through electrostriction (with the so-called q coefficient), such polarisation-induced stress being the one that acts upon the elastic dipole ? If not, where do the stresses acting on the elastic dipoles come from ?

The authors "propose that electric field-induced alignment of the dynamic elastic dipoles is the source of electrostriction." (Page 6). How do the authors envision that the electric field aligns the elastic dipoles (if not through the mechanism described above) ?

Beyond the coupling mechanism, if the electric dipoles and the elastic dipoles (that is, to me, what the authors call "the dopant-induced strain tensor per Zr ion") are coupled, then there should be a peak in the dielectric losses (related to a dielectric relaxation) at a frequency that is a multiple of the elastic relaxation related to the elastic dipoles. The symmetry of the structure imposes the relation between these two frequencies (see, e.g. *Anelastic Relaxation in Crystalline Solids* by Nowick and Berry, Academic Press, 1972). Has such joint relaxations been observed?

Having the dielectric and elastic relaxations related in such a way would convincingly evidence the coupling between the electric field and the elastic dipoles.

There is also the question of how this local anisotropic deformation leads to a macroscopic deformation. This remains unclear to me so far. The authors, on page 7, seem to indicate that the local deformation leads to a macroscopic deformation through a volume average. The authors state that if all the elastic dipoles were to align with the electric field, then the

deformation would be 60 times larger than the one measured. Why would all dipoles not align with the electric field? If it is because of the random orientation of the grains in the ceramics, then a geometric average should provide a more realistic value to be compared to the measured one.

A final note on this local-to-macro transition: according to the proposed explanation, the material should expand along the direction of the field and contract perpendicularly (at least this is what the dopant-induced strain tensor components seem to indicate). NCES have often been reported to be auxetic. Is it the case here? Why?

Technical questions:

What are the performances of non re-oxidized samples? The presence of Ce^{3+} implies the existence of oxygen vacancies. Do the samples (both for non-re-oxidized low Zr content and re-oxidized Zr content above 10mol%) exhibit electrostriction comparable to aliovalent substitutions (especially as a function of frequency)? This would help contrast the elastic dipoles created through aliovalent and isovalent substitutions.

How does the band gap change upon Zr substitution in the DFT calculations? Can this be related to the measured change in electric and mechanical properties through substitution, in addition or correlated to the reported stiffness of the CeO_8 vs ZrO_8 complexes? How does the density of state change, and does it resembles the difference observed e.g. between $GaBr_3$ and Ga_2O_3 reported in Trujillo et al. *npj Comp.Mat.* 8, 251 (2022)?

Have the authors evidenced any saturation of the strain or of the electrostrictive coefficient at higher fields? Is the induced polarisation linear with respect to the field up to the highest fields applied? This is important as electrostriction is only quadratic in electric field as long as the relationship between polarisation and field is linear, and the Q_h value calculated on page 7 is only valid if such linear relationship is evidenced.

On page 2, why is it advantageous not to develop polarisation under stress?

Fig1.a : why is Zr-doped samples absent?

Have the authors verified the linear relation between the applied stress and the variation of the relative permittivity? Eq.S5 agrees with Eq.S4 only if the trend is indeed linear. In addition, how was the dynamic stress applied? The photo only shows a vise that would enable a static force to be applied.

The numbering of the SI does not always correspond to the one used in the article (Inverse electrostriction measurements are in Section 6 and not 5 as indicated on page 3, the IS is in Section 5 and not 4 as indicated in the caption of Figure 3 and so on.). This should be easily fixed.

Fig.S8-1d: what is the unit of the supercell electric dipole moment? How does it compare with the polarisation measured on the whole sample? Is the ratio between the calculated and measured polarisation consistent with the one of the measured strain over the calculated one if all elastic dipoles were aligned with the electric field?

Table S8-3: what are the axes corresponding to the diagonalised dopant-induced strain tensor? What is the direction and amplitude of the Zr off-centering?

“NCES” is used on page 6 but only defined on page 7. Even though I too prefer “non-classical electrostrictors” to “giant” electrostrictors, clearly stating how the two are related in the introduction would improve the understanding of readers unfamiliar with the topic.

On page 7, the references to ionic conductors should also mention the work of Li et al. on LAMOX compounds (Phys.Rev.Mat. 2, 041403(R) 2018) to underline that, despite the obvious preeminence of this group on that topic, they are not the only ones to work on these materials.

Provided the requested clarifications about the nature of what the authors call an elastic dipole and how it is coupled to the electric field, I support the publication of this article in Nature Communications.

Reviewer #2 (Remarks to the Author):

Varenik et al. report the electrostrictive properties of Zr-doped Ceria ceramics (\leq Zr 20 %) which show large longitudinal electrostriction coefficient ($M_{33} \sim 10^{-16} \text{ m}^2/\text{V}^2$) and relatively small dielectric constant ($\epsilon_r \sim 200$). Authors claim that the large 2nd harmonic electromechanical coupling remains in the frequency range of a few tens mHz and a few kHz. The underlying mechanism on the observed large electromechanical coupling of $\text{Zr}_x\text{Ce}_{1-x}\text{O}_2$ is proposed by the cation size mismatch between smaller Zr dopant and the host Ce and the associated Zr motion in the fluorite cage (larger than Ce) under electric field applications. The inclusion and the local environment of Zr in the host CeO_2 lattices have been addressed by X-ray absorption spectroscopy, also supported by the theoretical calculations.

The experimentally obtained electrostriction over the measured frequency range ($\text{mHz} < f < \text{kHz}$) is large ($\sim 10^{-16} \text{ m}^2/\text{V}^2$) - similar large electrostriction were already observed in other doped ceria ceramics/films. I think the most important result for this work is the extension of large electrostriction towards higher frequency range (kHz). However, the experimental data (e.g. dielectric permittivity, electrical conductivity, and sample comparison) and interpretation are not in agreement with each other. For example, both dielectric permittivity and correlated electrostriction for the Zr-doped ceria are frequency-dependent. What is the mechanism behind the observed relaxation behaviours? Also, experimental results show that higher Zr doping in CeO_2 gives higher electrical conductivity and lower stiffness (lower Young's modulus). Why does the conductivity of the Zr-doped ceria ceramics increase with higher Zr doping? How can we completely avoid the internal/external effect (e.g. charged point defects) on the electromechanical coupling of the doped ceria ceramics? These make some of results very doubtful. Furthermore, the data presentation is poor and unclear and other important data are also missing (see my comments below). Therefore, I cannot recommend its publication in Nat. Commun.

These are my comments:

1. In pp 2, lines 30 – 31, the sentence “..., *suggesting a different mechanism for electrostriction.*” is vague. Authors should revise this sentence to be more specific.
2. The electrostriction of Zr-doped CeO_2 pellets is limited in the frequency range of < few

kHz (Fig. 2b). It looks like a just extended frequency range for the electrostriction of Zr-doped ceria, compared to the low-frequency electrostrictive response of aliovalent cation-doped CeO_{2-y} . Similar frequency dependence of dielectric relaxation behavior was also observed (Fig. S5). If the observed electrostriction is mainly due to the proposed dynamic motion of Zr in the ceria lattice cage, I expect it should extend to much higher frequency regime (like hard PZT, PMN, and PMN-PT). I wonder why it still shows a strong relaxation behavior for the electrostriction and permittivity in few kHz regimes. Authors should clarify this. Moreover, please add measurement data for the frequency-dependent (up to 1 MHz) 2nd harmonic response of a reference sample (e.g. PMN or PMN-PT) in Fig. 2b to compare and clarify the observed relaxation behavior of the doped ceria.

3. Have authors measured dielectric permittivity of the samples in different electric field conditions? It is assumed that the max. electric field of ~ 0.7 MV/m can be driven by applying a voltage of 700 V to a 1 mm-thick ceramic (Fig. 2a). Authors state that the dielectric permittivity of all the samples was measured by a small field (10 kV/m) (in the Method Section). For this case, only 10 V can be applied to a 1 mm-thick sample. I think that it is important to compare dielectric relaxation in different fields (/voltages) if possible. This would give a clearer picture for both the relaxation behaviors of the permittivity (/conductivity), and electrostriction of the samples (x).

4. What is the sign of the apparent electrostriction coefficients (either expansion or contraction) for the measured samples? Authors should define this rather putting the absolute values. It will be great to present the measured real-time 2nd harmonic displacements of the Zr-doped CeO_2 for readers. I guess that the authors can easily collect visible output responses (according to the high electrostriction coefficients, thus it should appear in micrometer scale) by using a laboratory oscilloscope.

5. Authors observed that higher Zr doping gives higher conductivity of $\text{Zr}_x\text{Ce}_{1-x}\text{O}_2$. Also, the real and imaginary parts of complex dielectric permittivity for $\text{Zr}_x\text{Ce}_{1-x}\text{O}_2$ ($x = 0, 0.05, 0.075, 0.1, 0.125, \text{ and } 0.2$) should be presented in the main text with sufficient information (i.e. voltage and frequency dependence, fitting equations and parameters).

6. Information on the paramagnetic behavior of $Zr_xCe_{1-x}O_2$ with and without V_o is relevant for this work? For my eyes, it is unnecessary... Instead, it will be much more important to show the electrostrictive performance of $Zr_xCe_{1-x}O_2$ with and without V_o inclusion. Please present the frequency dependence of the electrostriction of the reduced and oxidized $Zr_xCe_{1-x}O_2$ in either the main text or SI. I can see largely different dielectric relaxation behaviors of those samples as shown in Fig. S5b. I expect that the electrostriction of the reduced sample could be higher with further extended frequency regime if the derived real part of the permittivity is correct.

7. The host CeO_2 cubic structure remains with a large amount of Zr substitution although ZrO_2 has a monoclinic crystal structure at room temperature. The bond length and angle Zr-O (-4.7 %) and Zr-Ce (-1.5 %) in the host Ce cage are largely different with those of Ce-O and Ce-Ce. If we consider the distance ratio of cation-cation and cation-anion for a fluorite cubic structure, e.g. 1.645 for the cubic CeO_2 case, the distance ratio of Zr-O and Zr-Ce is ~ 1.7 . There should be a Zr-mediated local crystal distortion, so it could not be isotropic along all the crystallographic directions, especially for a monoclinic distortion case. However, it is a surprise for me that there is no local crystal symmetry change with a high Zr doping concentrations (up to 20 %). To verify this, authors should add at least the XRD 2theta profile data for the prepared/measured $Zr_xCe_{1-x}O_2$ ceramics as a function of x in Fig. S1. Further, what is the critical Zr doping concentration to maintain the cubic fluorite $Zr_xCe_{1-x}O_2$?

8. Lastly, in pp 8, lines 1-3, authors emphasize that large cation size mismatch between smaller dopant (Ca, Sr, Ba) and host Ce may deliver similar large electrostrictive effects. Please see a previous work (*Scripta Materialia*, 187, 183 (2020)) for a Ca-doped CeO_2 which is very unlikely the same effect with a relatively very low electrostriction coefficient of ~ 10 - $18 \text{ m}^2/\text{V}^2$... If authors want to generalize such a mismatch effect (with smaller cation dopants), M values of all the cases should be shown in the manuscript. Otherwise, it should be removed or revised properly.

Reviewer #3 (Remarks to the Author):

The manuscript of Varenik et al. entitled "Lead-free, low permittivity ceramics displaying giant electrostriction" presents exciting results. The results and findings are very important for the field, and thus should be published. As more research could lead to further improvements, this work opens new applications for electrostriction.

The large electrostrictive strain is indeed extraordinary for ceria that is doped with an isovalent ion, i.e., Zr⁴⁺ on the site of Ce⁴⁺. The results of 200 ppm strain at a field of 7 kV is only a factor of 2 short as compared to the champion relaxor PMN-PT (TRS technology web site), thus about equivalent. There is one large advantage over PMN-PT: CeZrO₂ exhibits a much lower dielectric constant, meaning that the driving electronics must deliver much less current, which is a technological advantage. Possibly also the temperature stability is better (not checked in the paper). If we compare with "standard" electrostrictive behaviour, which is applicable for densely packed material, we consider the fundamental relation of electrostriction $S=QD^2$ (S: strain, Q: electrostrictive coefficient, D: dielectric displacement field or polarisation P), it is plausible and experimentally well verified that Q is proportional to the compliance s and the inverse of the dielectric constant ϵ , i.e., $Q \propto s/\epsilon$ (Newnham). For the M coefficient as used in this paper, it follows that $S=Q*\epsilon^2E^2$, and $M \propto s*\epsilon$. The electrostriction of PMN is so large because the dielectric constant is huge (about 10'000). In CeZrO₂ this cannot apply. The dielectric constant is reported (in the paper) as only 250. In addition, the material is not so soft (no large compliance s). Its Young's modulus of 215 GPa is much larger than the one of PMN-PT and also of silicon (to have a comparison). Considering the basic material CeO₂ with an M of 10-19 m²/V² (see fig. 1b), and a dielectric constant of 30, we note that the increase of the dielectric constant and the softening of the structure by about 10 % would only lead to an increase of M of a factor of 10, but not of a factor of 1000! There is thus a factor of 100 to explain.

In recent years, there were a couple of papers presenting results about a giant electrostriction in Gd³⁺ doped ceria (CGO). In this case, a high density of oxygen vacancies is introduced in order to compensate the missing positive charges. The electric dipoles are readily identified in the form of $[(\text{Ce}^{\text{IV}}-\text{V}_\text{O})^\ominus]$, which have a preferential distance

(next-next neighbours). Oxygen vacancies align by hopping to an external electric field and lead to a huge dielectric constant at low frequencies, and to a giant electrostriction in the same frequency range (< 1 Hz) (see Park 2022, ref. 4 of the paper). Hence, the CGO case looks rather like the PMN-PT type with a huge dielectric constant.

The authors of this manuscript manage well to identify the reason for the additional contribution to the electrostriction: an elastic dipole around the point defect of Zr. Zr^{4+} is smaller than Ce^{4+} . The oxygen cube around Zr^{4+} is shrinking, leaving space for more motion of the ZrO_8 group. EXAFS and XANES spectroscopy show that the Zr-O bond is shorter than the Ce-O bond. There is only one value, and no change of orientation is observed. The cubic symmetry thus still prevails. DFT calculations show, however, that much less energy is needed to deform the oxygen cube around Zr, and even to displace the oxygen cube. The authors speak of a dynamic elastic dipole, i.e., one being easily formed upon an external force (stress or electric field). This explains the lowering of the stiffness. Since the elastic dipole consists of ions, it is also clear that an electric field can also deform the ZrO_8 group. In fact, what is needed is an electric dipole of the ZrO_8 group and a coupling with the elastic dipole. In perfect cubic symmetry (and absence of oxygen vacancies or Ce^{3+} ions) there would be no such dipole. This seems to be the situation at zero electric field. As the electric field displaces every atom individually in insulators, the ZrO_8 group is distorted directly by an electric field, and leads as a consequence to the formation of an elastic dipole. Apparently, the available space leads to large distortion of the cube, which explains the observed behaviour. The authors call this a dynamic elastic dipole. Is it understood that this is the consequence of the electric field? If yes, it should be mentioned earlier in the paper, and in the abstract.

Other remarks:

The statement on p.2: "In aliovalent doped ceria (CeO_2), an electrostriction strain coefficient as large as that of PMNPT ($10^{-16} m^2/V^2$) coexists with relative permittivity of < 30 " is not correct. There is a large increase above this value. When M is increasing, the permittivity increases as well. One should measure both values at the same frequency, and not at different ones. In the cited ref. 11, the increase of the permittivity is well reported. This sentence should be changed. The increase of the dielectric constant (or the D-field) is

also reported in ref. 4.

The induced strain is negative? In fig. 1b a compressive strain is mentioned. As a negative strain is the consequence of a compressive stress (which is not present in this case) one could conclude that a negative strain is meant.

REVIEWER COMMENTS

Reviewer #1 (Remarks to the Author):

Reviewer #1 Comment 1:

So called “giant electrostrictors” have so far suffered from meagre responses at frequencies above 100Hz. As such their ability to be integrated in actuators is limited. Electrostriction coefficient M of the order of $10^{16} \text{ m}^2/\text{V}^2$ is at par with relaxor ferroelectrics but still below what has been reported e.g. for hybrid perovskites and composites. In addition, the developed strain is also of the same order of magnitude than PMN-PT. Therefore, the use of “giant electrostriction” in the title seems not to be appropriate. The advantage of Zr-doped CeO₂ compared to the composites is the stress developed thanks to their superior mechanical properties. Such advantage may be mentioned to underline the potential of this material in efficient actuators and as a credible alternative to PMN-PT.

Reviewer #1 Response 1:

We thank the reviewer for raising this issue. We consider Zr-doped ceria to be a "giant" electrostrictor because the values of M_{33} (and Q_h) reported for this material are two orders of magnitude larger than those calculated on the basis of Newnham's scaling law. Recently, the term “giant electrostriction” has been re-analyzed and a strict definition provided¹. In the manuscript, we adhered to this definition and, for clarity, have revised the text in three locations:

a) In the Introduction section on page 2:

These ceramics joined other groups of materials, such as hybrid perovskites^{2,3} and polymer composites^{4,5}, which have been classified¹ as “giant” electrostrictors” by virtue of displaying an electrostriction coefficient that is at least one order of magnitude larger than that predicted by Newnham's scaling law.

b) In the Results section on page 8:

The value of $|Q_h|$, the hydrostatic electrostriction polarization coefficient, predicted for a classical electrostrictor with the relative dielectric permittivity and elastic modulus of $\text{Zr}_{0.1}\text{Ce}_{0.9}\text{O}_2$ ^{6,7} is $\approx 0.07 \text{ m}^4/\text{C}^2$. This is more than two orders of magnitude lower than the actual measured value, $9.7 \text{ m}^4/\text{C}^2$, thereby identifying Zr-doped ceria as a non-classical electrostrictor (NCES).

c) In the Introduction section on page 2, where the advantages of Zr-doped ceria electrostrictors are presented:

In contrast to PMN-PT, doped ceria is non-toxic and fully compatible with Si-based microfabrication processes^{8,9}, has a high elastic modulus, as well as low dielectric permittivity. All three factors favorably distinguish ceria-based electrostrictors from other electrostrictive materials currently in use.

Reviewer #1 Comment 2:

The first main point to clarify is what an elastic dipole is for the authors. According to Kroner (Kröner, E. (1958). "Kontinuumstheorie der Versetzungen und Eigenspannungen," Springer, Berlin.) it is a point defect producing local distortions that react to a homogenous stress applied to the material in an analogous way an electric dipole reacts to an applied electric field (albeit one is a second-rank tensor and the other one a vector). As such, the strain tensor induced by the shorter Zr-O bonds (within the preserved fluorite structure) constitutes an elastic dipole. Hence, there should be what the authors call "equilibrium" elastic dipoles, even in the absence of an applied electric field. Would the authors define what an elastic dipole is for them ?

Would the authors explain why they state there is no equilibrium elastic dipole ?

I recognise that the nature of the defect causing the elastic dipole is different from the one involving an oxygen vacancy but, to me, both lead to a local deformation susceptible to give rise to an elastic dipole.

Reviewer #1 Response 2:

We thank the reviewer for raising this issue. We do adhere to the definition of elastic dipoles given by Kroner. However, for *"producing local distortions that react to a homogenous stress applied to the material in a manner that is analogous to the way an electric dipole reacts to an applied electric field"*, the symmetry of the local elastic field of the defect would have to be lower than the average symmetry of the lattice. Otherwise, the elastic dipoles would not experience a driving force to reorient. The importance of this requirement is discussed in detail by Arthur Nowick in Ref 10 below (which is also Ref. 29 in the revised manuscript). This is also the approach followed by the DFT calculations. There is insufficient space to elaborate on these calculations in the main text, where we only stated that "The elastic dipole tensor for Zr as dopant, including both local and long range deformation, was also computed using DFT. ^{11, 12}". The computational details are provided in SI Section 8.2. Quoting in brief, "the short-range elastic dipole tensor \mathbf{G} can be associated with $E_{Zr,u}^f$ (Zr substitutional defect formation energy) for a given applied strain tensor through the first order term in a Taylor expansion, (Eq. S8.2) $E_{Zr,u}^f = E_{Zr,u=0}^f + \mathbf{G}:\mathbf{u}$ ", where $E_{Zr,u}^f$ is the defect formation energy for a given strain state, \mathbf{u} ; and \mathbf{G} is the elastic dipole tensor. After obtaining \mathbf{G} , the dopant-induced strain tensor per Zr ion (α_C), can be written as (Eq. S 8.4), $\alpha_C = -\frac{(\mathbb{C}^{-1}\mathbf{G})}{V_U}$, where \mathbb{C} is the elastic stiffness tensor and V_U is the volume per formula unit of CeO₂. Any point defect can cause local bond distortion, which will be balanced by the long-range elastic field.

In fully oxidized Zr-doped ceria, the concentration of oxygen vacancies is comparable to known values for ceria under ambient conditions; neither the EXAFS data nor the theoretical DFT modeling reveal deviation from cubic symmetry around the Zr ions which substitute for Ce ions within the host lattice. Consequently, the local elastic field around the Zr ions must be different from that induced electrostatically by oxygen vacancies. On the other hand, the application of an electric field generates uniaxial strain, which requires an asymmetric elastic field. Thus, for the case of Zr-doped ceria, we suggest that the elastic dipoles are produced by the electric field. Therefore, these elastic dipoles are called "dynamic" to distinguish them from those that break local symmetry even in the absence of anisotropic mechanical stress or applied electric field. We have used the terms "dynamic" vs. "static" elastic dipoles in parts of the original manuscript. In the revised manuscript, we have now replaced all use of the term "equilibrium" dipoles with "static" dipoles.

The revised manuscript now includes two additional clarifying statements in the Introduction section on page 3:

Elastic dipoles are defined as an anisotropic elastic field, capable of reorientation under external anisotropic stress¹⁰. In the case of aliovalent-doped ceria, elastic dipoles are present even in the absence of an external electric field. Once applied, the field is able to reorient these pre-existing, "static" elastic dipoles.

and

In contrast to aliovalent doped ceria, X-ray absorption spectroscopy (XAS) and theoretical DFT modeling do not find local deviation from cubic symmetry in the vicinity of the Zr ions substituting for Ce ions within the host lattice. This eliminates the possibility of pre-existing "static" elastic dipoles associated with the Zr dopant; rather, the local elastic field becomes anisotropic only under external field application due to the anharmonicity of the bonds of the eight oxygen ions which are near neighbors to the Zr ion.

Reviewer #1 Comment 3:

Fig2.c shows a linear relationship between the log of the high frequency M_{33} and the Zr doping content up to 10mol%. How do the authors relate this evolution to the number of elastic dipoles ? Why is exponential and not linear?

Reviewer #1 Response 3:

We are grateful to the reviewer for asking this question. The reviewer is correct in noting that M_{33} increases exponentially with Zr-concentration, which implies some cooperative effect. Currently, neither XAS data nor DFT calculations provide insight into the origin of this effect and it will have to remain a subject for future studies. In response to the reviewer's comment, the following sentence was added on page 4 to the text of the revised manuscript:

The longitudinal electrostriction strain coefficient increases exponentially with Zr-content in the 0-10 mol% concentration range.

Reviewer #1 Comment 4:

The question of the coupling mechanism to the electric field also remains to be clarified.

Reviewer #1 Response 4:

We appreciate this comment as it has helped us improve the manuscript by further clarifying the nature of the coupling mechanism. Electrostriction must arise from the anharmonicity of chemical bonds (perfectly harmonic bonds do not present even-order response). For the case of Zr-doped ceria, the anharmonicity is related to the $[\text{ZrO}_8]$ units in the CeO_2 host lattice. As suggested by AIMD simulations, the electric field easily polarizes these units, modifying local bond lengths (see Figure 6b), which in turn generates mechanical strain.

We have added the following clarifying text in the Discussion section on page 8:

Since $[\text{ZrO}_8]$ units are easily polarizable (see Figure 6b), the external electric field displaces them from the equilibrium position, generating a macroscopic mechanical strain.

Reviewer #1 Comment 5:

As an elastic dipole is defined by its reaction to a mechanical stress. Do the authors consider that the electric field induces a polarisation that, in turns, induces a stress through electrostriction (with the so-called q coefficient), such polarisation-induced stress being the one that acts upon the elastic dipole ? If not, where do the stress acting on the elastic dipoles come from ?

Reviewer #1 Response 5:

This question is addressed in our response to Comment 4. Indeed, as the reviewer has correctly summarized the sequence of events: the electric field displaces $[\text{ZrO}_8]$ -units with anharmonic bonds to a greater extent than the host $[\text{CeO}_8]$ units. This generates local mechanical strain, which in turn produces macroscopic strain derived from strain additivity. The additional text, introduced in response to the previous comment, clarifies this issue as well.

Reviewer #1 Comment 6:

The authors “propose that electric field-induced alignment of the dynamic elastic dipoles is the source of electrostriction.” (Page 6). How do the authors envision that the electric field align the elastic dipoles (if not through the mechanism described above)?

Reviewer #1 Response 6:

The reviewer is correct. Based on the combined experimental results and theoretical simulations, we propose this mechanism. The electric field displaces the elastic dipoles in the direction of the field, thereby generating macroscopic strain due to strain additivity (see also our reply to comment 5).

Reviewer #1 Comment 7:

Beyond the coupling mechanism, if the electric dipoles and the elastic dipoles (that is, to me, what the authors call "the dopant-induced strain tensor per Zr ion") are coupled, then there should be a peak in the dielectric losses (related to a dielectric relaxation) at a frequency that is a multiple of the elastic relaxation related to the elastic dipoles. The symmetry of the structure imposes the relation between these two frequencies (see, e.g. Anelastic Relaxation in Crystalline Solids by Nowick and Berry, Academic Press, 1972).

Has such joint relaxations been observed?

Having the dielectric and elastic relaxations related in such a way would convincingly evidence the coupling between the electric field and the elastic dipoles.

Reviewer #1 Response 7:

We agree with the reviewer. Unfortunately, our dynamic mechanical analyzer (as is true for most other DMA instruments) does not function at frequencies above 100 Hz. Therefore, we cannot measure the mechanical (elastic) relaxation directly. However, the presence of a dielectric loss peak is consistent with the expectation of the reviewer (see below). In response to the reviewer's comment, a graph was added to the supplementary material as Figure S5-3. The revised main text now references this graph.

Figure S5-3. Room temperature loss tangent for oxidized, 10mol%Zr-doped ceria pellets, 0 VDC bias, 10 VAC, frequency range 1 MHz-100Hz, measured with stainless steel electrodes of which the upper one was spring-loaded. Impedance measurements were made in the same device as the converse electrostriction measurements.

Additional text in the Results section on page 4 that references this graph:

A dielectric loss peak is also observed within this range of frequencies (Fig. S5-3).

Reviewer #1 Comment 8:

There is also the question of how this local anisotropic deformation leads to a macroscopic deformation. This remains unclear to me so far. The authors, on page 7, seem to indicate that the local deformation leads to a macroscopic deformation through a volume average. The authors state that if all the elastic dipoles were to align with the electric field, then the deformation would be 60 times larger than the one measured. Why would all dipoles not align with the electric field? If it is because of the random orientation of the grains in the ceramics, then a geometric average should provide a more realistic value to be compared to the measured one.

Reviewer #1 Response 8:

Elastic fields are additive. Therefore, microscopic strain may cooperate in producing macroscopic strain when the elastic dipoles acquire a preferred orientation. This is the principle of volume average cited in the manuscript on page 7. Within the complete range of electric field strengths investigated (≤ 13.4 kV/cm), the quadratic dependence of macroscopic strain on electric field is preserved in experiments. Therefore, strain saturation of elastic dipoles has not been reached. Our current model cannot predict the magnitude of the displacement of the Zr ion. Therefore, an estimate of the upper bound for the elastic dipole is provided. The reviewer is correct in stating that the maximum strain estimated from the modeling is more than 60 times larger than the measured value, which is what would be expected. Random orientation of grains may also contribute to theory overestimation.

The text on pages 7-8 relating to this question was rephrased:

This is ~60 times larger than the electrostrictive strain measured with the E-field < 13.4 kV/cm used here. This value is consistent with our estimate of an upper bound on Zr displacement. Since no strain saturation is observed in Figure 2a, we suggest that not all elastic dipoles are aligned under the field.

Reviewer #1 Comment 9:

A final note on this local-to-macro transition: according to the proposed explanation, the material should expand along the direction of the field and contract perpendicularly (at least this is what the dopant-induced strain tensor components seem to indicate). NCEs have often been reported to be auxetic. Is it the case here? Why?

Reviewer #1 Response 9:

We appreciate this comment. Electrostriction is known to produce both expansion and contraction. According to R. Newnham^{6, 13} it is related to the nature of the chemical bond anharmonicity. For instance, most perovskites elongate along the field, while a single crystal of fluorite CaF₂ contracts along the field ($Q_{11} < 0$,¹⁴). We are not aware of any relationship between Poisson's ratio and the longitudinal electrostriction strain coefficient. Poisson's ratio for a number of doped ceria ceramics has been measured and it is consistently positive. It is also known that most oxide non-classical electrostrictors (e.g., aliovalent-doped ceria, delta-phase Bi₂O₃, nominally dry and hydrated acceptor-doped BaZrO₃) contract along the field. We have added additional clarifying text in the manuscript in order to discuss this point as raised by the reviewer.

The following changes were made in the text on page 4:

The ceramic samples studied in this work, contract along the field, *i.e.*, both parallel and antiparallel (Figure S9), similar to previously reported doped ceria^{15, 16}, delta-phase Bi₂O₃⁷, nominally dry and hydrated acceptor-doped BaZrO₃¹⁷, and fluorides⁶, despite displaying Poisson's ratio > 0 (*i.e.* none are auxetic).

Reviewer #1 Comment 10:

Technical questions:

What are the performances of non re-oxidized samples? The presence of Ce³⁺ implies the existence of oxygen vacancies. Do the samples (both for non-re-oxidized low Zr content and re-oxidized Zr content above 10mol%) exhibit electrostriction comparable to aliovalent substitutions (especially as a function of frequency)? This would help contrast the elastic dipoles created through aliovalent and isovalent substitutions.

Reviewer #1 Response 10:

Based on the SQUID measurements presented in supplementary section 4, the concentration of Ce³⁺ in the re-oxidized 10 mol% Zr samples is ~ 100ppm, and in the reduced (non-re-oxidized) samples is ~400 ppm. The additional oxygen vacancies required for charge compensation are sufficient to reduce the longitudinal electrostriction strain coefficient by more than an order of magnitude. Since the behavior is complex and beyond the scope of this work, we are preparing a separate manuscript which addresses this specific question.

Reviewer #1 Comment 11:

How does the band gap change upon Zr substitution in the DFT calculations? Can this be related to the measured change in electric and mechanical properties through substitution, in addition or correlated to the reported stiffness of the CeO₈ vs ZrO₈ complexes? How does the density of state change, and does it resembles the difference observed e.g. between GaBr₃ and Ga₂O₃ reported in Trujillo et al. *npj Comp.Mat.* 8, 251 (2022)?

Reviewer #1 Response 11:

We thank the reviewer for the insightful question. Here we present the calculated partial density of states (PDOS) for CeO₂ and Zr-doped-CeO₂ (with doping rate 1/32). The PDOS is shown in Fig R1. Overall, replacing Ce with Zr does not significantly alter the electronic properties, similar to those reported for unreduced Zr-doped CeO₂ in the literature^{18, 19, 20}.

Fig R-1, The computed partial density of states (PDOS) for (a) bulk CeO₂ and (b) 3.125% Zr doped CeO₂.

Fig R1 shows the PDOS plotted with respect to the Fermi level, which coexists with the Valence Band Maximum. The experimentally measured band gap is predominantly oxygen 2p with respect to the conduction band of Ce-5d²¹ This gap is 5.17 eV in Fig R1 and does not change upon Zr-doping.

The localized Ce-4f band is within the gap. The energy difference between the occupied O-2p bands (in range [-3.93, 0] eV) and the unoccupied Ce-4f band (in range [2.16, 3.03] eV) remains at around 2.16 eV after Zr-doping. Some calculations reported the gap between O-2p and Ce-4f as the band gap. With this definition, the band does not change upon Zr-doping.

A new state belonging to unoccupied Zr-4d orbital appeared at ~4.50 eV above the Fermi-level. In bulk ZrO₂, the conduction band is mainly derived from Zr 4d orbitals. The very localized Zr-4d peak at ~4.50 eV will have limited contributions to electron conduction. The rest of Zr-4d band is mixed with Ce-5d orbitals. Therefore, we conclude that localized Zr-4d orbitals will contribute to a reduced band gap and increase the dielectric permittivity, which is inversely proportional to the band gap²², likely very locally.

Trujillo et al.²³ applied Newnham's semi-empirical relationship to DFT computed stress-dependent dielectric constants to search for high electrostrictive materials based on bulk DFT calculations. They noticed that the shape of the DOS (e.g., GaBr₃ and Ga₂O₃), i.e., a more spread DOS vs. DOS with narrow peaks, can be correlated to the mechanical properties. We did not observe such a difference upon Zr-doping, consistent with the similar mechanical modulus measured experimentally.

In response to the reviewer's comments:

We have added the PDOS plot to the SI as Fig. S8-3 and have added the following to the main text on page 5:

While the increase in the relative permittivity is consistent with the appearance of a localized Zr-4d electronic state within the CeO₂ band gap, predominantly between O-2p and Ce-5d orbitals, the decrease in elastic modulus cannot be correlated with any electronic structure change²³ (Fig S8-3).

Reviewer #1 Comment 12:

Have the authors evidenced any saturation of the strain or of the electrostrictive coefficient at higher fields? Is the induced polarisation linear with respect to the field up to the highest fields applied? This is important as electrostriction is only quadratic in electric field as long as the relationship between polarisation and field is linear, and the Qh value calculated on page 7 is only valid if such linear relationship is evidenced.

Reviewer #1 Response 12:

This work used relatively low electric fields (<13.4 kV/cm) for measurement of the direct electrostriction effect and observed a linear dependence of strain on the field squared up to 225 ppm (Fig. 2a). This suggests that a) the measured values of strain are far from saturation; and b) the material remains a linear dielectric within the whole range of electric fields.

The text was modified in the following way on page 4:

The longitudinal electrostrictive strain remains linear with the applied electric field squared within the accessible range of fields (0-13.4 kV/cm). This indicates that the strain achieved (225 ppm) is far from saturation and that the material remains within the linear dielectric regime (i.e., polarization remains linearly proportional to the electric field).

Reviewer #1 Comment 13:

On page 2, why is it advantageous not to develop polarisation under stress ?

Reviewer #1 Response 13:

From the practical point of view, the fact that electrostrictors do not develop polarization under stress is definitely an advantage since, in its absence, the shock wave generated by a transducer will not interfere with the driving circuit of the transducer itself. Such a problem exists for all piezoelectric transducers and requires special measures to maintain the required shape of the generated sonic wave (pulse).

We have made the following change in the Introduction on page 2 in response to this constructive comment of the reviewer:

as the former do not develop polarization under stress allowing for higher actuation accuracy and simpler driving electrical circuitry.

Reviewer #1 Comment 14:

Fig1.a : why is Zr-doped samples absent ?

Reviewer #1 Response 14:

For the re-oxidized Zr samples, there is no difference between the low and high frequency electrostriction strain coefficients (*cf.*, Fig2b).

Reviewer #1 Comment 15:

Have the authors verified the linear relation between the applied stress and the variation of the relative permittivity? Eq.S5 agrees with Eq.S4 only if the trend is indeed linear. In addition, how was the dynamic stress applied? The photo only shows a vise that would enable a static force to be applied.

Reviewer #1 Response 15:

- a) A figure presenting the dependence of the real component of the relative dielectric permittivity on applied static compressive stress, and for AC frequencies between 10^1 - 10^6 Hz, was added to the supplementary information (see below).

Figure S6 2. Increase in the relative real component of the dielectric permittivity ϵ' under uniaxial compressive stress (S), for an oxidized 10 mol% Zr-doped ceria ceramic pellet. ϵ_0 is the dielectric permittivity of vacuum.

This figure was also referenced in the main text in the Results section on page 4 as follows:

The change in dielectric permittivity remained linear within the measured range of applied compressive stress (Figure S6-2).

Reviewer #1 Comment 16:

The numbering of the SI does not always correspond to the one used in the article (Inverse electrostriction measurements are in Section 6 and not 5 as indicated on page 3, the IS is in Section 5 and not 4 as indicated in the caption of Figure 3 and so on.). This should be easily fixed.

Reviewer #1 Response 16:

Fixed

Reviewer #1 Comment 17:

Fig.S8-1d: what is the unit of the supercell electric dipole moment? How does it compare with the polarisation measured on the whole sample? Is the ratio between the calculated and measured polarisation consistent with the one of the measured strain over the calculated one if all elastic dipoles were aligned with the electric field?

Reviewer #1 Response 17:

Fig.S8-1d, Fig.6b and Eq. S8.7 share the same electric dipole unit debye (D), $1D \approx 0.2082 e \cdot \text{\AA}$. We added the unit definition in the revised figure captions and also in the main text for clarification.

Fig.S8-1d shows the time evolution of the supercell electric dipole moment generated by room temperature AIMD with the NVT ensemble and in the absence of external polarization or strain in order to show that no significant electric dipole emerges if the system is not perturbed. This is not intended to be compared with the polarization measured on the whole sample, which contains many Zr-dopants.

The dynamic elastic dipole associated with each Zr-dopant was calculated in Fig 6. We have made an estimate of the net strain it can cause if all the elastic dipoles are aligned. In the original manuscript (Page 7) we stated:

We propose that electric field-induced alignment of the dynamic elastic dipoles is the source of electrostriction. In a sample of 10 mol% Zr-doped ceria with initially randomly oriented dynamic elastic dipoles, complete alignment of all the elastic dipoles parallel to the applied electric field or uniaxial mechanical stress may generate strain of $\approx 0.1 \cdot 0.145 = 1.45\%$ or 14500 ppm. This is ~ 60 times larger than the electrostrictive strain measured with E-field < 13.4 kV/cm used here, consistent with our upper bound displacement estimation and also suggesting that not all the elastic dipoles are aligned under the field.

This text has now been modified in accordance with the reviewer's comments (pages7-8):

We propose that electric field-induced alignment of the dynamic elastic dipoles, is the source of the increased electrostriction strain coefficient. In addition, the dielectric relaxation observed in the range of a few kHz is due to the inability of the induced dipole to follow the alternating field. In a sample of 10 mol% Zr-doped ceria with dynamic elastic dipoles, complete alignment of all the elastic dipoles parallel to the applied electric field, or to uniaxial mechanical stress, may generate strain of $\approx 0.1 \cdot 0.145 = 1.45\%$ or 14500 ppm. This is ~ 60 times larger than the electrostrictive strain measured with the E-field < 13.4 kV/cm used here. This is consistent with our estimate of an upper bound on Zr displacement. Since no strain saturation is observed in **Error! Reference source not found.**, we suggest that not all elastic dipoles are aligned under the field.

Reviewer #1 Comment 18:

Table S8-3: what are the axes corresponding to the diagonalised dopant-induced strain tensor? What is the direction and amplitude of the Zr off-centering?

Reviewer #1 Response 18:

We thank the reviewer for this question. By diagonalization of the dopant-induced strain tensor per Zr ion α_C , we obtained the anisotropy of the strain tensor projected onto the principal directions.

$$\mathbf{P}^{-1}\alpha_C\mathbf{P} = \alpha_{C,P} = \begin{bmatrix} \alpha_{C,P,1} & 0 & 0 \\ 0 & \alpha_{C,P,2} & 0 \\ 0 & 0 & \alpha_{C,P,3} \end{bmatrix}$$

Writing \mathbf{P} as a block matrix of its column vectors β_i : $\mathbf{P} = [\beta_1, \beta_2, \beta_3]$. They are the eigenvectors of α_C that satisfy $\alpha_C\beta_i = \alpha_{C,P,i}\beta_i$, and they are the principal directions as well. For example, for the Zr-off centered 0.22 Å along the [111] direction, the calculated principal directions are: [-0.578, -0.578, -0.576], [-0.696, 0.718, -0.022] and [-0.426, -0.388, 0.817], which are close to $[\bar{1} \ \bar{1} \ \bar{1}]$, $[\bar{1} \ 1 \ 0]$, $[\bar{1} \ \bar{1} \ 2]$.

In the revised manuscript, we added the eigenvectors of the diagonalized α_C . The direction and amplitude of the Zr off-centering are specified as well.

specified as well.

Table S8-3 The computed elastic dipole tensor \mathbf{G} and the dopant-induced strain tensor per Zr ion, $\alpha_{\mathbf{C}}$, for different Zr structures. The diagonalization of $\alpha_{\mathbf{C}}$ gives 3 eigenvectors, which are the principal directions onto which the strain tensor $\alpha_{\mathbf{C}}$ is projected.

	\mathbf{G}			$\alpha_{\mathbf{C}}$		
	in Cartesian coordinates	in Cartesian coordinates	Diagonalized	Eigenvectors		
Fully relaxed Zr at the Ce-center	$\begin{bmatrix} 9.358 & -0.001 & -0.001 \\ -0.001 & 9.358 & -0.001 \\ -0.001 & -0.001 & 9.358 \end{bmatrix}$	$\begin{bmatrix} -0.066 & 0 & 0 \\ 0 & -0.066 & 0 \\ 0 & 0 & -0.066 \end{bmatrix}$	$\begin{bmatrix} -0.066 & 0 & 0 \\ 0 & -0.066 & 0 \\ 0 & 0 & -0.066 \end{bmatrix}$	$\begin{bmatrix} 1 & 0 & 0 \\ 0 & 1 & 0 \\ 0 & 0 & 1 \end{bmatrix}$		
Snapshot from AIMD 300K 3500fs	$\begin{bmatrix} -3.539 & -0.089 & -0.089 \\ -0.089 & -3.154 & 1.017 \\ -0.089 & 1.017 & -3.038 \end{bmatrix}$	$\begin{bmatrix} 0.028 & 0.006 & 0.064 \\ 0.006 & 0.021 & -0.073 \\ 0.064 & -0.073 & 0.019 \end{bmatrix}$	$\begin{bmatrix} -0.078 & 0 & 0 \\ 0 & 0.031 & 0 \\ 0 & 0 & 0.115 \end{bmatrix}$	$\begin{bmatrix} 0.457 & -0.544, & -0.704 \\ -0.750 & -0.661 & 0.024 \\ 0.478 & -0.517 & 0.710 \end{bmatrix}$		
Snapshot from AIMD 300K 3900fs	$\begin{bmatrix} -5.712 & -1.308 & 0.474 \\ 1.308 & -5.479 & 1.806 \\ 0.474 & 1.806 & -5.276 \end{bmatrix}$	$\begin{bmatrix} 0.042 & 0.093 & -0.034 \\ 0.093 & 0.038 & -0.129 \\ -0.034 & -0.129 & 0.035 \end{bmatrix}$	$\begin{bmatrix} 0.214 & 0 & 0 \\ 0 & 0.008 & 0 \\ 0 & 0 & -0.107 \end{bmatrix}$	$\begin{bmatrix} -0.475 & -0.670 & 0.571 \\ -0.817 & 0.094 & -0.569 \\ -0.328 & 0.737 & 0.592 \end{bmatrix}$		
Zr-off center 0.22 Å along [111] direction	$\begin{bmatrix} 7.645 & -0.938 & -0.925 \\ -0.938 & 7.645 & -0.925 \\ -0.925 & -0.925 & 7.640 \end{bmatrix}$	$\begin{bmatrix} -0.054 & 0.067 & 0.066 \\ 0.067 & -0.054 & 0.066 \\ 0.066 & 0.066 & -0.054 \end{bmatrix}$	$\begin{bmatrix} 0.079 & 0 & 0 \\ 0 & -0.121 & 0 \\ 0 & 0 & -0.119 \end{bmatrix}$	$\begin{bmatrix} -0.578 & -0.578 & -0.576 \\ -0.696 & 0.718 & -0.022 \\ -0.426 & -0.388 & 0.817 \end{bmatrix}$		

Reviewer #1 Comment 19:

“NCES” is used on page 6 but only defined on page 7. Even though I too prefer “non-classical electrostrictors” to “giant” electrostrictors, clearly stating how the two are related in the introduction would improve the understanding of readers unfamiliar with the topic.

Reviewer #1 Response 18:

We have added the necessary definitions in the Introduction section, as mentioned in our response to comment 1.

Reviewer #1 Comment 20:

On page 7, the references to ionic conductors should also mention the work of Li et al. on LAMOX compounds (Phys.Rev.Mat. 2, 041403(R) 2018) to underline that, despite the obvious preeminence of this group on that topic, they are not the only ones to work on these materials.

Reviewer #1 Response 20:

The reference was added (49).

Reviewer #1 Comment 21:

Provided the requested clarifications about the nature of what the authors call an elastic dipole and how it is coupled to the electric field, I support the publication of this article in Nature Communications.

Reviewer #1 Response 21:

Details concerning the nature of the elastic dipoles are included in our response to reviewer #1- comments 4 and 7.

Reviewer #2 (Remarks to the Author):

Reviewer #2 Comment 1:

Varenik et al. report the electrostrictive properties of Zr-doped Ceria ceramics (\leq Zr 20 %) which show large longitudinal electrostriction coefficient ($M_{33} \sim 10^{-16} \text{ m}^2/\text{V}^2$) and relatively small dielectric constant ($\epsilon_r \sim 200$). Authors claim that the large 2nd harmonic electromechanical coupling remains in the frequency range of a few tens mHz and a few kHz. The underlying mechanism on the observed large electromechanical coupling of $\text{Zr}_x\text{Ce}_{1-x}\text{O}_2$ is proposed by the cation size mismatch between smaller Zr dopant and the host Ce and the associated Zr motion in the fluorite cage (larger than Ce) under electric field applications. The inclusion and the local environment of Zr in the host CeO_2 lattices have been addressed by X-ray absorption spectroscopy, also supported by the theoretical calculations. The experimentally obtained electrostriction over the measured frequency range ($\text{mHz} < f < \text{kHz}$) is large ($\sim 10^{-16} \text{ m}^2/\text{V}^2$) - similar large electrostriction were already observed in other doped ceria ceramics/films. I think the most important result for this work is the extension of large electrostriction towards higher frequency range (kHz). However, the experimental data (e.g. dielectric permittivity, electrical conductivity, and sample comparison) and interpretation are not in agreement with each other.

Reviewer #2 Response 1:

We trust that our replies, as detailed below, will satisfactorily address all the comments of the Reviewer.

Reviewer #2 Comment 2:

For example, both dielectric permittivity and correlated electrostriction for the Zr-doped ceria are frequency-dependent.

What is the mechanism behind the observed relaxation behaviours?

Also, experimental results show that higher Zr doping in CeO₂ gives higher electrical conductivity and lower stiffness (lower Young's modulus). Why does the conductivity of the Zr-doped ceria ceramics increase with higher Zr doping?

Reviewer #2 Response 2:

1. According to the proposed mechanism consistent with XAS data and DFT calculations, electrostriction in Zr-doped ceria is generated by field-induced displacement of the polarizable elastic dipoles related to [ZrO₈] units. The observed relaxation in dielectric permittivity (Fig.S5.3 in the revised version), as well as in the electromechanical strain response, is a manifestation of the characteristic response time of the [ZrO₈] units. Dielectric and mechanical relaxation in electrostrictors with large M_{33} is well known, viz., the most common commercial electrostrictor PMN-PT. In response to the reviewer's comment, a statement that relaxation is related to the characteristic response time of [ZrO₈] units has been added to the text on page 7:

In addition, the dielectric relaxation observed in the frequency range of a few kHz is due to the inability of the induced elastic/electric dipole to follow the alternating electric field.

2. The source of the increase in total conductivity under ambient conditions, with increase in Zr doping, is difficult to assign. Even the very well-studied Gd-doped ceria ceramics display a not completely understood increase in intermediate temperature conductivity until approx. 10 mol% doping and then the conductivity decreases. At room temperature, ion conductivity is expected to be negligible while electron conductivity is likely present. So it is possible that with increased Zr doping, the observed increase in Ce³⁺ concentration although small, may provide hopping electron (polaron) transport in ceria.

In reply to the reviewer's comment, the following text has been introduced into the revised manuscript on page 5:

The source of the increase in the total measured conductivity as a function of Zr dopant concentration is difficult to assign. It may be due, at least in part, to a small increase in the concentration of Ce³⁺, which will increase the number of electrons in the conduction band.

Reviewer #2 Comment 3:

How can we completely avoid the internal/external effect (e.g. charged point defects) on the electromechanical coupling of the doped ceria ceramics? These make some of results very doubtful. Furthermore, the data presentation is poor and unclear and other important data are also missing (see my comments below). Therefore, I cannot recommend its publication in Nat. Commun.

Reviewer #2 Response 3:

Since the reviewer does not detail the type of “internal/external effect (e.g. charged point defects)” that may influence the electromechanical coupling in doped ceria ceramics and does not offer a possible alternative interpretation of our results, we describe below two common effects that may influence the electromechanical measurements:

1. Joule heating can of course lead to thermal expansion, which may be mistaken for a second-order mechanical response. However, at all frequencies examined, the Zr-doped ceria pellets contract under the applied electric field, with negative strain ≤ 225 ppm. For comparison, we note that the thermal expansion coefficient of ceria is ~ 11 ppm/K.
2. Charge trapping and electret-like electromechanical response may be observed in good dielectrics, e.g., polymers, glasses,²⁴ where the electron relaxation time is very long. This is not the case for our samples: **(a)** electrets respond at the first harmonic of the AC field (as they display remanent polarization^{24, 25}), whereas Zr-doped ceria respond only at the second harmonic. **(b)** electret strain relaxes with time constant shorter than the dielectric relaxation time of the sample. The dielectric relaxation time for Zr-doped ceria calculated from our data is $10^7 \text{ Ohm}\cdot\text{m} \times 220 \epsilon_0 \approx 20 \text{ ms}$ (50 Hz)! Had trapped charges influenced our measurements, the dependence of M_{33} on field should have displayed relaxation between 0 Hz and >1 kHz.

We would also like to point out that our elastic modulus measurements, along with the >200 ppm strain generated, predict stress of > 48 MPa. We are not aware of any mechanism other than non-classical electrostriction that is able to generate strain and stress of such magnitude in Zr-doped CeO_2 .

Along with the reviewer, we are keenly interested in finding an alternative mechanism that could compete with electrostriction and, in so doing, may cast doubt upon the interpretation of our results. To date, we haven't found one. We trust that our experimental data and theoretical modelling will inspire members of the materials science, physics, and chemistry community to work to verify and/or fine-tune the proposed mechanism. In response to the comments of all the reviewers, we have worked to improve our data presentation and have added clarifying comments and additional information.

Reviewer #2 Comment 4:

These are my comments:

1. In pp 2, lines 30 – 31, the sentence “..., suggesting a different mechanism for electrication.” is vague. Authors should revise this sentence to be more specific.

Reviewer #2 Response 4:

We thank the reviewer for this comment. The sentence has now been modified to read (page 3):

Electrostrictive strain at higher frequencies (*i. e.*, M_{33}^{100} Hz) increases exponentially with decreasing dopant radius ¹⁵ (**Error! Reference source not found.**b), suggesting an additional mechanism for electrostriction, one which is independent of dopant valence and consequently does not require the presence of oxygen vacancies.

Reviewer #2 Comment 5:

2. The electrostriction of Zr-doped CeO₂ pellets is limited in the frequency range of < few kHz (Fig. 2b). It looks like a just extended frequency range for the electrostriction of Zr-doped ceria, compared to the low-frequency electrostrictive response of aliovalent cation-doped CeO_{2-y}. Similar frequency dependence of dielectric relaxation behavior was also observed (Fig. S5). If the observed electrostriction is mainly due to the proposed dynamic motion of Zr in the ceria lattice cage, I expect it should extend to much higher frequency regime (like hard PZT, PMN, and PMN-PT). I wonder why it still shows a strong relaxation behavior for the electrostriction and permittivity in few kHz regimes. Authors should clarify this. Moreover, please add measurement data for the frequency-dependent (up to 1 MHz) 2nd harmonic response of a reference sample (e.g. PMN or PMN-PT) in Fig. 2b to compare and clarify the observed relaxation behavior of the doped ceria.

Reviewer #2 Response 5:

The mechanisms of electrostriction in Zr-doped ceria and in PMN-based electrostrictors are profoundly different: while the strains that they are able to generate are similar in magnitude (hundreds of ppm) they are opposite in sign. According to XAS data and theoretical modeling, ZrO₈-units are stiff and are responsible for the increase in relative dielectric permittivity (to ~225 for 10mol% Zr). This should be compared with PMN or PMN-PT, where displacement of the B-site ion within the oxygen octahedron is responsible for increase in permittivity ($\epsilon > 2000$).

Another way to visualize the difference is to consider the diffuse X-ray scattering²⁶, which is exceptionally high for PMN-based materials and non-existent for Zr-doped ceria. In the former, the lattice is strongly disordered, while in the latter, only the ZrO₈-units generate strain, i.e., the remainder of the lattice is largely undisturbed. It is worth noting that the frequency relaxation for PMN-PT single crystals, with $\leq 33\text{mol\% PbTiO}_3$, is still in the 100 kHz range (see for instance²⁷), decreasing with reduction in PbTiO₃ content, and increasing the relative dielectric permittivity to tens of thousands. From this point of view, the fact that our material provides comparable M_{33} and strain, but with $\epsilon \sim 225$, presents striking practical advantage.

We have added the following clarifying text on page 8 and have added Fig.S11:

The relaxation frequency is also somewhat lower than expected for the dynamic motion of a cation (such as in PMN-PT, Figure S11). This may perhaps be attributed to the involvement of the oxygen coordination shell.

Figure S11. Electrostriction strain coefficient in PMN-PT15 (TRS ceramics), at 0.2 kV/cm. A 50% reduction is observed at 19.4kHz. Thermal expansion is detected close to relaxation due to the increase in the imaginary component of the dielectric permittivity (ϵ''), thereby causing Joule heating. Below the relaxation frequency, the real component of the dielectric permittivity exceeds 10,000, which presents serious practical difficulties.

Reviewer #2 Comment 6:

3. Have authors measured dielectric permittivity of the samples in different electric field conditions? It is assumed that the max. electric field of ~ 0.7 MV/m can be driven by applying a voltage of 700 V to a 1 mm-thick ceramic (Fig. 2a). Authors state that the dielectric permittivity of all the samples was measured by a small field (10 kV/m) (in the Method Section). For this case, only 10 V can be applied to a 1 mm-thick sample. I think that it is important to compare dielectric relaxation in different fields (/voltages) if possible. This would give a clearer picture for both the relaxation behaviors of the permittivity (/conductivity), and electrostriction of the samples (x).

Reviewer #2 Response 6:

We have responded to a similar question **(12) of Reviewer 1**. However, for the convenience of Reviewer #2, we respond here as well.

In this report, a relatively low electric field (< 13.4 kV/cm) was used to measure the direct electrostriction strain: we observed linear dependence of the strain on the field squared up to 225 ppm (Fig. 2a; N.B. we have replaced E_{pp}^2 by E^2). This suggests that a) the measured values of the strain are far from saturation; and b) the material remains a linear dielectric within the range explored.

The text on page 4 was modified in the following way:

The longitudinal electrostrictive strain remains linear with the applied electric field squared within the accessible range of fields (0-13.4 kV/cm). This indicates that the strain achieved (225 ppm) is far from saturation and that the material remains within the linear dielectric regime (i.e., polarization remains linearly proportional to the electric field).

We note that the equipment which we have available is not capable of applying both high excitation voltage and high frequency (maxima: 10kHz, at 4kV). To achieve a field of 0.7^* MV/m with 10 V excitation voltage would require the sample to be less than 14 μ m thick. This is very likely to cause ceramic breakage.

Reviewer #2 Comment 7:

4. What is the sign of the apparent electrostriction coefficients (either expansion or contraction) for the measured samples? Authors should define this rather putting the absolute values. It will be great to present the measured real-time 2nd harmonic displacements of the Zr-doped CeO₂ for readers. I guess that the authors can easily collect visible output responses (according to the high electrostriction coefficients, thus it should appear in micrometer scale) by using a laboratory oscilloscope.

Reviewer #2 Response 7:

The original manuscript stated that all samples contract in the direction of the field (Figure 1, legend). This statement is now also included in the Results section on pages 3-4:

All samples described in this work contract along the field ($M_{33} < 0$), irrespective of the field direction (i.e. parallel or anti-parallel) (Figure S9), similar to previously reported aliovalent doped ceria^{15, 16}, delta-phase Bi₂O₃⁷, nominally dry or hydrated acceptor-doped BaZrO₃¹⁷, and fluoride minerals⁶. In fact, none are auxetic, meaning all display positive Poisson's ratio.

Following the reviewer's comment, a real time displacement-time trace superimposed on the applied voltage has now been added to the supplementary material (see below):

Figure S9. Real time strain and voltage measurements during an electrostriction experiment at $f = 0.15$ Hz, $V = 1550$ Volt, 10 mol% Zr doped ceria pellet.

Reviewer #2 Comment 8:

5. Authors observed that higher Zr doping gives higher conductivity of $Zr_xCe_{1-x}O_2$. Also, the real and imaginary parts of complex dielectric permittivity for $Zr_xCe_{1-x}O_2$ ($x = 0, 0.05, 0.075, 0.1, 0.125, \text{ and } 0.2$) should be presented in the main text with sufficient information (i.e. voltage and frequency dependence, fitting equations and parameters).

Reviewer #2 Response 8:

In the revised version of the manuscript, we present these data for $x = 0.05, 0.1$ (OX) in the supplementary material (S5). As stated in the original manuscript, Zr-contents of $x=0.125, 0.2$ are too high to suppress cerium reduction. Therefore, these data are not included.

Reviewer #2 Comment 9:

6. Information on the paramagnetic behavior of $Zr_xCe_{1-x}O_2$ with and without V_o is relevant for this work? For my eyes, it is unnecessary... Instead, it will be much more important to show the electrostrictive performance of $Zr_xCe_{1-x}O_2$ with and without V_o inclusion. Please present the frequency dependence of the electrostriction of the reduced and oxidized $Zr_xCe_{1-x}O_2$ in either the main text or SI. I can see largely different dielectric relaxation behaviors of those samples as shown in Fig. S5b. I expect that the electrostriction of the reduced sample could be higher with further extended frequency regime if the derived real part of the permittivity is correct.

Reviewer #2 Response 9:

We appreciate these suggestions of the reviewer. We believe however that showing the results obtained for samples that are not fully oxidized will not provide a valid comparison; the concentration of V_o cannot be reliably controlled in the absence of aliovalent dopants, as is usually done for ceria. Therefore, we include only one example (Figure S10) in the supplementary material of the revised manuscript. This figure supports our claim concerning the detrimental effect of oxygen vacancies on the magnitude of electrostrictive strain in Zr-doped ceria.

With regard to the SQUID measurements: to the best of our knowledge, they are the only way to assess the content of Ce^{3+} and, thereby, of V_o . We are therefore confident that the data presented are indeed relevant.

Reviewer #2 Comment 10:

7. The host CeO₂ cubic structure remains with a large amount of Zr substitution although ZrO₂ has a monoclinic crystal structure at room temperature. The bond length and angle Zr-O (-4.7 %) and Zr-Ce (-1.5 %) in the host Ce cage are largely different with those of Ce-O and Ce-Ce. If we consider the distance ratio of cation-cation and cation-anion for a fluorite cubic structure, e.g. 1.645 for the cubic CeO₂ case, the distance ratio of Zr-O and Zr-Ce is ~1.7. There should be a Zr-mediated local crystal distortion, so it could not be isotropic along all the crystallographic directions, especially for a monoclinic distortion case. However, it is a surprise for me that there is no local crystal symmetry change with a high Zr doping concentrations (up to 20 %). To verify this, authors should add at least the XRD 2theta profile data for the prepared/measured Zr_xCe_{1-x}O₂ ceramics as a function of x in Fig. S1. Further, what is the critical Zr doping concentration to maintain the cubic fluorite Zr_xCe_{1-x}O₂?

Reviewer #2 Response 10:

In response to the reviewer's comment, we have added the XRD patterns for $x \leq 0.2$ to the supplementary material as Figure S1. No diffraction peaks other than those due to the fluorite phase are visible. This is not surprising ; ZrO₂-CeO₂ solid solutions have been thoroughly studied in the past due to strong interest on the part of the catalyst community²⁸. Consistent with the XRD data, the Zr K-edge XANES spectrum for $x=0.1$ (Fig 5) supports a "quasi"-cubic [ZrO₈] structure.

The ZrO₂-CeO₂ phase diagram (see below, from the NIST database or²⁹) indeed predicts mixed phases at room temperature. However, following application of the rapid sintering protocol²⁷ (specifically developed to prevent cation diffusion) to the co-precipitated powders, cation diffusion in ceria is so slow that upon cooling to room temperature, it does not occur. Therefore, the cubic structure is kinetically stable.

[FIGURE REDACTED]

Reviewer #2 Comment 11:

8. Lastly, in pp 8, lines 1-3, authors emphasize that large cation size mismatch between smaller dopant (Ca, Sr, Ba) and host Ce may deliver similar large electrostrictive effects. Please see a previous work (Scripta Materialia, 187, 183 (2020)) for a Ca-doped CeO₂ which is very unlikely the same effect with a relatively very low electrostriction coefficient of ~10-18 m²/V²... If authors want to generalize such a mismatch effect (with smaller cation dopants), M values of all the cases should be shown in the manuscript. Otherwise, it should be removed or revised properly.

Reviewer #2 Response 11:

Our manuscript focuses on the influence of isovalent dopants (Zr) in fluorite structured ceria on the magnitude of electrostrictive strain. Our note in the text cites “cation size mismatch between smaller, isovalent dopants (e.g., Ca, Sr, Ba)” in minerals with a fluorite structure: CaF₂, SrF₂, BaF₂; the ratio of crystal radii Ca/Sr or Sr/Ba is close to that of Zr/Ce. Aliovalent Ca-doping in CeO₂ is therefore not relevant to our discussion. Electrostriction in aliovalent-doped ceria has been extensively reported (^{15, 30}) and, indeed, the measured values of |M₃₃| may vary from 10⁻¹⁸ m²/V² for Ca-doped ceria to ~ 10⁻¹⁶ m²/V² for Gd- and Sm- doped ceria at frequencies <1Hz.

We have emphasized this point in the text on pages 8-9:

In fact, this novel NCES mechanism may be more generally applicable. A small dopant cation, along with nearest neighbor anions, vibrating anharmonically in a relatively large (compared to the host cation) cage may facilitate electromechanical coupling. Since the square faces of [MO₈] are less stiff than the triangular faces of an octahedron in perovskites or a tetrahedron in sphalerite, a fluorite lattice may be particularly suited to large dopant-host cation size mismatch.

Reviewer #3 (Remarks to the Author):

Reviewer #3 Comment 1:

The manuscript of Varenik et al. entitled "Lead-free, low permittivity ceramics displaying giant electrostriction" presents exciting results. The results and findings are very important for the field, and thus should be published. As more research could lead to further improvements, this work opens new applications for electrostriction.

Reviewer #3 Response 1:

We thank the reviewer for his favorable assessment of our findings.

Reviewer #3 Comment 2:

The large electrostrictive strain is indeed extraordinary for ceria that is doped with an isovalent ion, i.e., Zr⁴⁺ on the site of Ce⁴⁺. The results of 200 ppm strain at a field of 7 kV is only a factor of 2 short as compared to the champion relaxor PMN-PT (TRS technology web site), thus about equivalent. There is one large advantage over PMN-PT: CeZrO₂ exhibits a much lower dielectric constant, meaning that the driving electronics must deliver much less current, which is a technological advantage. Possibly also the temperature stability is better (not checked in the paper). If we compare with "standard" electrostrictive behaviour, which is applicable for densely packed material, we consider the fundamental relation of electrostriction $S=QD^2$ (S : strain, Q : electrostrictive coefficient, D : dielectric displacement field or polarisation P), it is plausible and experimentally well verified that Q is proportional to the compliance s and the inverse of the dielectric constant ϵ , i.e., $Q \propto s/\epsilon$ (Newnham). For the M coefficient as used in this paper, it follows that $S=Q\epsilon^2E^2$, and $M \propto s*\epsilon$. The electrostriction of PMN is so large because the dielectric constant is huge (about 10'000). In CeZrO₂ this cannot apply. The dielectric constant is reported (in the paper) as only 250. In addition, the material is not so soft (no large compliance s). Its Young's modulus of 215 GPa is much larger than the one of PMN-PT and also of silicon (to have a comparison).*

Considering the basic material CeO₂ with an M of 10-19 m²/V² (see fig. 1b), and a dielectric constant of 30, we note that the increase of the dielectric constant and the softening of the structure by about 10 % would only lead to an increase of M of a factor of 10, but not of a factor of 1000! There is thus a factor of 100 to explain.

Reviewer #3 Response 2:

We appreciate the reviewer's comment. While non-classical electrostriction is still a very young field, we note that Newnham's scaling law, mentioned above, has been validated for standard or "classical" electrostrictors. It is that fact that renders the material we describe so very interesting. The mechanism we propose for Zr-doped ceria attributes its non-classical behavior to the presence of the dynamic elastic dipoles of the [ZrO₈]-units.

The text on page 2 was modified to emphasize this point:

The combination of low dielectric permittivity and high elastic modulus places the value of $|Q_h|$ for doped ceria at least two orders of magnitude above the prediction by Newnham's scaling law^{6, 7} and identifies it, as well as other recently described ceramics with a large concentration of point defects^{7, 17, 31, 32, 33, 34}, as "giant", or non-classical electrostrictors^{1, 6}. These ceramics have joined other groups of materials, such as hybrid perovskites^{2, 3} and polymer composites^{4, 5}, classified¹ as "giant" electrostrictors", by virtue of displaying an electrostriction coefficient that is at least one order of magnitude larger than that predicted by Newnham's scaling law.

Reviewer #3 Comment 3:

In recent years, there were a couple of papers presenting results about a giant electrostriction in Gd³⁺ doped ceria (CGO). In this case, a high density of oxygen vacancies is introduced in order to compensate the missing positive charges. The electric dipoles are readily identified in the form of $[\text{Cd}]_{\text{Ce}}\text{-V}_{\text{O}}$, which have a preferential distance (next-next neighbours). Oxygen vacancies align by hopping to an external electric field and lead to a huge dielectric constant at low frequencies, and to a giant electrostriction in the same frequency range (< 1 Hz) (see Park 2022, ref. 4 of the paper). Hence, the CGO case looks rather like the PMN-PT type with a huge dielectric constant.

The authors of this manuscript manage well to identify the reason for the additional contribution to the electrostriction: an elastic dipole around the point defect of Zr. Zr⁴⁺ is smaller than Ce⁴⁺. The oxygen cube around Zr⁴⁺ is shrinking, leaving space for more motion of the ZrO₈ group. EXAFS and XANES spectroscopy show that the Zr-O bond is shorter than the Ce-O bond. There is only one value, and no change of orientation is observed. The cubic symmetry thus still prevails. DFT calculations show, however, that much less energy is needed to deform the oxygen cube around Zr, and even to displace the oxygen cube. The authors speak of a dynamic elastic dipole, i.e., one being easily formed upon an external force (stress or electric field). This explains the lowering of the stiffness. Since the elastic dipole consists of ions, it is also clear that an electric field can also deform the ZrO₈ group. In fact, what is needed is an electric dipole of the ZrO₈ group and a coupling with the elastic dipole. In perfect cubic symmetry (and absence of oxygen vacancies or Ce³⁺ ions) there would be no such dipole. This seems to be the situation at zero electric field. As the electric field displaces every atom individually in insulators, the ZrO₈ group is distorted directly by an electric field, and leads as a consequence to the formation of an elastic dipole. Apparently, the available space leads to large distortion of the cube, which explains the observed behaviour. The authors call this a dynamic elastic dipole. Is it understood that this is the consequence of the electric field? If yes, it should be mentioned earlier in the paper, and in the abstract.

Reviewer #3 Response 3:

The Reviewer's concise summary of our findings is accurate. As requested, we have added the following text to the revised manuscript.

In the Abstract:

Unlike the elastic dipoles reported for aliovalent doped ceria, which are present even in the absence of an applied elastic or electric field, the elastic dipoles in Zr_xCe_{1-x}O₂ are formed only under applied anisotropic field.

And in the Introduction on page 3:

In contrast to aliovalent doped ceria, X-ray absorption spectroscopy (XAS) and theoretical DFT modeling do not find local deviation from cubic symmetry in the vicinity of the Zr ions within the host ceria lattice. This eliminates the possibility of pre-existing “static” elastic dipoles associated with the Zr dopant. Rather, due to bond anharmonicity, the local elastic field becomes anisotropic only upon application of an external field. Zr-O bonds were found to be shorter by ~0.1 Å than Ce-O bonds and highly anharmonic, due to the expanded range of motion available for [ZrO₈] local bonding units compared to the [CeO₈] host. These conditions give rise to “dynamic” elastic dipoles, i.e., elastic dipoles that are formed only under an external field due to anharmonicity, revealing a previously unknown mechanism of non-classical electrostriction.

Other remarks:

Reviewer #3 Comment 4:

The statement on p.2: “In aliovalent doped ceria (CeO₂), an electrostriction strain coefficient as large as that of PMNPT (10-16 m²/V²) coexists with relative permittivity of < 30” is not correct. There is a large increase above this value. When M is increasing, the permittivity increases as well. One should measure both values at the same frequency, and not at different ones. In the cited ref. 11, the increase of the permittivity is well reported. This sentence should be changed. The increase of the dielectric constant (or the D-field) is also reported in ref. 4.

Reviewer #3 Response 4:

We agree with the reviewer that the dielectric permittivity should be measured at the same frequency as M. However, this requirement poses difficulties at frequencies <10 Hz for aliovalent doped ceria with high ionic conductivity (Gd, Sm, Nd). Attempting to measure dielectric permittivity at these frequencies polarizes both the electrodes and the grain boundaries. The result appears as an “apparent” increase in the dielectric permittivity below 10 Hz at low fields (Ref. 11 cited above) and, if the field is sufficiently high, (e.g., in very thin films of Gd-doped ceria – ref 4.) even at much higher frequencies. Therefore, it is not possible to determine what the dielectric permittivity of these materials may be at a few Hz. However, for intermediate frequencies (5-10 Hz) where Gd, Sm, or Nd-doped ceria ceramics do undergo strain relaxation, M_{33} is still well above 10^{-17} m²/V² and the “apparent” relative dielectric permittivity remains below 100, rather than tens of thousands expected for a classical electrostrictor with comparable M_{33} and elastic modulus. When Lu or Yb are the dopants, there is no reduction in $M_{33} \approx 10^{-17}$ m²/V² until approx. 100 Hz (^{15,15}), and the value of the relative dielectric permittivity <30 in the frequency range 1-100 Hz is not in doubt. We have therefore modified the text on page 2 in order to avoid confusion.

For some compositions of aliovalent doped ceria, e.g. Lu- or Yb- doped ceria, the longitudinal electrostriction strain coefficient is $\approx 10^{-17}$ m²/V², which is only approximately a factor of 10 below that of PMN-PT, the best electrostrictor currently in use ^{35, 36}. However, between 1-100 Hz, M_{33} for the doped ceria ceramics coexists with low relative dielectric permittivity $\epsilon < 30$ ¹⁶ (vs. $\epsilon_{PMN-PT} > 10,000$) and elastic modulus more than twice as large as that of PMN-PT.

Reviewer #3 Comment 5:

The induced strain is negative? In fig. 1b a compressive strain is mentioned. As a negative strain is the consequence of a compressive stress (which is not present in this case) one could conclude that a negative strain is meant.

Reviewer #3 Response 5:

Indeed, the macroscopic strains measured are negative – this is seen both as negative deformation in the direct electrostriction measurements, and as reverse behavior of the electrostriction coefficient in converse measurements (increase of dielectric constant with increase in compressive stress). This is a common (although still not well explained) phenomenon in fluorite crystals (e.g., CaF_2)¹⁴

We have introduced clarifying text, both in the on-line Methods section, and in the caption to Figure 1b:

All samples contract parallel to the applied field.

In addition, the y-axis label in Figure 2a now reads "Compressive (i.e. negative) strain (ppm)".

References for the response letter

1. Yu J. C., Janolin P. E., Defining "giant" electrostriction. *J. Appl. Phys.* **131**, (2022).
2. Chen B., *et al.*, Large electrostrictive response in lead halide perovskites. *Nat Mater* **17**, 1020+ (2018).
3. Gao Z. R., *et al.*, Ferroelectricity of the Orthorhombic and Tetragonal MAPbBr₃ Single Crystal. *J Phys Chem Lett* **10**, 2522+ (2019).
4. Yuan J. K., *et al.*, Giant Electrostriction of Soft Nanocomposites Based on Liquid Crystalline Graphene. *ACS Nano* **12**, 1688-1695 (2018).
5. Luna A., *et al.*, Giant Electrostrictive Response and Piezoresistivity of Emulsion Templated Nanocomposites. *Langmuir* **33**, 4528-4536 (2017).
6. Newnham R., Sundar V., Yimnirun R., Su J., Zhang Q., Electrostriction: nonlinear electromechanical coupling in solid dielectrics. *The Journal of Physical Chemistry B* **101**, 10141-10150 (1997).
7. Yavo N., *et al.*, Large Nonclassical Electrostriction in (Y, Nb)-Stabilized delta-Bi₂O₃. *Adv. Funct. Mater.* **26**, 1138-1142 (2016).
8. Santucci S., Zhang H., Sanna S., Pryds N., Esposito V., Electro-chemo-mechanical effect in Gd-doped ceria thin films with a controlled orientation. *J. Mater. Chem. A* **8**, 14023-14030 (2020).
9. Makagon E., *et al.*, All-Solid-State Electro-Chemo-Mechanical Actuator Operating at Room Temperature. *Adv. Funct. Mater.* **31**, 2006712 (2021).
10. Nowick A. S., Berry B. S. *Anelastic relaxation in crystalline solids*. Academic Press: New York, 1972.
11. Leslie M., Gillan M. J., The Energy and Elastic Dipole Tensor of Defects in Ionic-Crystals Calculated by the Supercell Method. *J. Phys. C: Solid State Phys.* **18**, 973-982 (1985).
12. Das T., Nicholas J. D., Sheldon B. W., Qi Y., Anisotropic chemical strain in cubic ceria due to oxygen-vacancy-induced elastic dipoles. *Phys. Chem. Chem. Phys.* **20**, 15293-15299 (2018).
13. Nomura S., Tonooka K., Kuwata J., Cross L. E., Newnham R. E., Electrostriction in Pb(Mg_{1/3}Nb_{2/3})O₃ Ceramics. *Ferroelectrics* **29**, 124-124 (1980).
14. Sundar V., Newnham R. E., Converse method measurements of electrostriction coefficients in low-K dielectrics. *Mater Res Bull* **31**, 545-554 (1996).
15. Varenik M., *et al.*, Trivalent Dopant Size Influences Electrostrictive Strain in Ceria Solid Solutions. *ACS Appl. Mater. Interfaces* (2021).
16. Kabir A., Tinti V. B., Varenik M., Lubomirsky I., Esposito V., Electromechanical dopant-defect interaction in acceptor-doped ceria. *Mater. Adv.* **1**, 2717-2720 (2020).

17. Makagon E., Kraynis O., Merkle R., Maier J., Lubomirsky I., Non-Classical Electrostriction in Hydrated Acceptor Doped BaZrO₃: Proton Trapping and Dopant Size Effect. *Adv. Funct. Mater.* **31**, Artn 2104188 (2021).
18. Muhich C., Steinfeld A., Principles of doping ceria for the solar thermochemical redox splitting of H₂O and CO₂. *J. Mater. Chem. A* **5**, 15578-15590 (2017).
19. Yang Z., Woo T. K., Hermansson K., Effects of Zr doping on stoichiometric and reduced ceria: A first-principles study. *J Chem Phys* **124**, (2006).
20. Waseem S., *et al.*, Structural, electronic and optical study of Zr:CeO₂ thin films by computational and experimental approach. *Physica B: Condensed Matter* **653**, 414671 (2023).
21. Schmitt R., *et al.*, A review of defect structure and chemistry in ceria and its solid solutions. *Chem. Soc. Rev.* **49**, 554-592 (2020).
22. Wang B. H., *et al.*, High-k Gate Dielectrics for Emerging Flexible and Stretchable Electronics. *Chem Rev* **118**, 5690-5754 (2018).
23. Trujillo D. P., *et al.*, Data-driven methods for discovery of next-generation electrostrictive materials. *Npj Comput Mater* **8**, (2022).
24. Kestelman V. N., Pinchuk L. S., Goldade V. A. *Electrets in engineering: fundamentals and applications*. Springer Science & Business Media, 2000.
25. Wada N., *et al.*, Fundamental electrical properties of ceramic electrets. *Mater Res Bull* **48**, 3854-3859 (2013).
26. You H., Zhang Q. M., Diffuse x-ray scattering study of lead magnesium niobate single crystals. *Phys. Rev. Lett.* **79**, 3950-3953 (1997).
27. Lebrun L., *et al.*, Investigations on ferroelectric PMN-PT and PZN-PT single crystals ability for power or resonant actuators. *Ultrasonics* **42**, 501-505 (2004).
28. Devaiah D., Reddy L. H., Park S. E., Reddy B. M., Ceria-zirconia mixed oxides: Synthetic methods and applications. *Catal Rev* **60**, 177-277 (2018).
29. Li L., *et al.*, Estimation of the phase diagram for the ZrO₂-Y₂O₃-CeO₂ system. *J Eur Ceram Soc* **21**, 2903-2910 (2001).
30. Kabir A., *et al.*, Electro-chemo-mechanical properties in nanostructured Ca-doped ceria (CDC) by field assisted sintering. *Scr. Mater.* **187**, 183-187 (2020).
31. Santucci S., Zhang H. W., Sanna S., Pryds N., Esposito V., Electro-chemo-mechanical effect in Gd-doped ceria thin films with a controlled orientation. *J. Mater. Chem. A* **8**, 14023-14030 (2020).
32. Santucci S., *et al.*, Electromechanically active pair dynamics in a Gd-doped ceria single crystal. *Phys. Chem. Chem. Phys.* **23**, 11233-11239 (2021).

33. Tinti V. B., Kabir A., Han J. K., Molin S., Esposito V., Gigantic electro-chemo-mechanical properties of nanostructured praseodymium doped ceria. *Nanoscale* **13**, 7583-7589 (2021).
34. Han J., *et al.*, Enhanced electromechanical properties in low-temperature gadolinium-doped ceria composites with low-dimensional carbon allotropes. *J. Mater. Chem. A* **10**, 4024-4031 (2022).
35. Kabir A., Bowen J. R., Varenik M., Lubomirsky I., Esposito V., Enhanced Electromechanical Response in Sm and Nd Co-doped Ceria. *Materialia* **12**, 100728 (2020).
36. Varenik M., *et al.*, Dopant Concentration Controls Quasi-Static Electrostrictive Strain Response of Ceria Ceramics. *ACS Appl. Mater. Interfaces* **12**, 39381-39387 (2020).

REVIEWER COMMENTS

Reviewer #1 (Remarks to the Author):

The authors have addressed all the points mentioned in my report. I hope they helped improve the manuscript that can now be published.

Reviewer #2 (Remarks to the Author):

The authors have been responsive to some of my comments very well and improved some parts of the manuscript significantly. The main discussion on the generation of large electrostriction in Zr-doped Ceria (\leq Zr 12.5%) remains the same. I noticed that in the revised manuscript the authors attempted to make the results in a line by excluding certain data presented in the previous version of the manuscript, e.g., the conductivity (at $100 \text{ mHz} < f < \text{MHz}$) and permittivity (at $100 \text{ Hz} < f < \text{MHz}$) of the oxidized $\text{Zr}_{0.1}\text{Ce}_{0.9}\text{O}_2$ and reduced $\text{Zr}_{0.1}\text{Ce}_{0.9}\text{O}_{2-x}$, measured at $100 \text{ mHz} < f < \text{MHz}$ (in Fig. S5-2). The proposed mechanism behind the observed electrostriction has become much clearer now thanks to other reviewers' help. It is indeed an interesting finding that the large electrostrictive properties (effective $M_{33}^* \sim 10^{-16} \text{ m}^2/\text{V}^2$) can dominantly occur in cubic fluorite CeO_2 structure by incorporating isovalent dopants smaller than Ce^{4+} , without a significant contribution of oxygen vacancies. In particular, the large electrostriction of $\text{Zr}_{0.1}\text{Ce}_{0.9}\text{O}_2$ is extended up to a few kHz unlike large electrostriction in aliovalent cation doped CeO_{2-x} , usually limited by $f < 10 \text{ Hz}$. I agree now that this work may deliver a significant material development on fluorite-based electrostrictors scientifically and technologically if it is non-classical as the authors just claim.

I should mention that the electromechanical properties of the just named non-classical/giant electrostrictors are not completely established yet. The electromechanical properties of non-classical/nonlinear electrostrictors are mostly based on the motion of charged point defects/ions. Thus, the situation is different with $\text{Zr}_{0.1}\text{Ce}_{0.9}\text{O}_2$ [stiff ZrO_8 (no increase in M at low frequency), the absence of mobile (/dynamical) defects, and no static elastic dipoles], compared to other non-classical electrostrictors (e.g., aliovalent cation-doped CeO_{2-x}). It is still an intriguing puzzle that $\text{Zr}_{0.1}\text{Ce}_{0.9}\text{O}_2$ also shows a small permittivity (experimental), but a large electrostriction.

To clarify this, I (also other two reviewers) asked the apparent permittivity of the $Zr_xCe_{1-x}O_2$ samples under the same electric fields, which were applied to the electromechanical measurements of the samples (although the authors present the converse effect now). The authors could not measure the permittivity of the samples in the same field conditions (due to their technical issue), applied for their electromechanical measurements. Also, in the authors' reply, some of my last comments (e.g., the conductivity/imaginary part of complex dielectric constants in frequency as a function of Zr%) were just ignored. They are all important for readers!

In summary, the research finding reported here is interesting and important to the related research community. However, I recommend that the authors address the points below before considering publication.

1. It is fine to provide a perspective on the impact of cation size mismatch in the discussion/summary section. However, the title of the manuscript is too general. Since the authors investigated only one material system, it should be specified with $Zr_xCe_{1-x}O_2$ in the title.
2. In my view, the electrostrictive strain doesn't linearly increase with E^2 in Fig. 2a although the newly updated Fig. S6-2 shows linearity with the converse effect. The authors used a large symbol size for plotting the data in Fig. 2a. Please confirm this with error bars. If not perfectly linear, the word, "linear" should be changed as to "almost linear". That is why I previously asked about the appearance of non-linearity with additional effect (e.g., oxygen vacancies, also see a paramagnetic response in the oxidized Zr10% sample). Can the authors completely rule out the oxygen vacancy effect from their experiments to deliver a concrete conclusion? Also, if the Zr doped ceria samples have an electrically nonlinear nature, what can we expect from?
3. The authors addressed it with a limit of dipolar switching (dielectric relaxation) of ZrO_8 at high frequencies (a few kHz). After that, the conductivity ($1/\epsilon''$) of the samples increases. I can understand that it is challenging to assign the conducting source in the samples with Zr% (e.g., an increased structural/lattice distortion, defect formation and electronic

contribution). However, to emphasize and manifest the Zr doping effect in ceria, the authors should present the frequency dependent M33 of Zr-doped ceria samples as a function of Zr% in the frequency range of $100 \text{ mHz (or } 10 \text{ Hz)} < f < 1 \text{ MHz}$ and incorporate them in Fig. 2b. I am guessing the damping (/relaxation) frequency of M increases up to Zr = 12.5 % and then it will decrease after 12.5 %. This variation can be compared with the frequency-dependent conductivity variation with Zr%.

4. In Fig. S9 of the revised manuscript, there is a phase lag for the 2nd harmonic displacement response of a $\text{Zr}_{0.1}\text{Ce}_{0.9}\text{O}_2$ sample in response to the applied AC voltage in time. What is its origin? Does the phase lag vary with frequency? Also, how long does the mechanical response of the electrically excited sample last (endurance)? I strongly suggest including long-time measurement data, either in the main text or in SI, to further emphasize the large and sustainable electrostriction of the developed ceria. Also, plot a strain Vs applied voltage curve and describe the origin of the phase lag in the figure caption.

5. In Fig. S10 of the revised manuscript, the M33 of the reduced sample in the frequency range of $100 \text{ mHz} - 100 \text{ Hz}$ shows much lower than that of the oxidized one. But, in Fig. S10 of the first version of the manuscript, the conductivity of the reduced and oxidized samples is very similar in the measured frequency range. Also, in the first version, the dielectric permittivity of the reduced sample at $f = 100 \text{ Hz}$ is larger than that of the oxidized one. The data are not consistent! Why don't they agree with each other? This is very strange to me.

6. The SQUID data shows that there is still a paramagnetic-like response even in the oxidized sample. What is the origin of this paramagnetism in the sample? Is it because of the presence of oxygen vacancies in an irregular manner? If so, both the oxidized and reduced samples possess oxygen vacancies with different amounts. The authors just mention a known defect concentration in SI. The valence state of Ce ions can also be checked by EELS and XAS (Ce L-edge spectra, for example, Wu et al., *Physica Scripta*. T115, 802 (2005)). Have the authors measured the XAS Ce L-edge spectra of the undoped and doped ceria samples?

7. Lastly, all the ceramic samples studied here are polycrystalline, as shown in XRD. This means that grains oriented along the (111), (200), (220), and (311) crystallographic

directions can be excited under uniaxial electric field applications. The authors claim that the motion of ZrO₈ is the key to generating large electromechanical displacement (negative longitudinal electrostrictive strain) in Zr doped ceria. In Fig. 6d, there are positive and negative displacements in the ZrO₈ with different crystallographic directions with a positive Poisson's ratio. In the experiment, all grain orientations cannot be aligned by electric field. I wonder how Zr/O vibrations in differently oriented grains respond to electric field application and how they generate the corresponding uniaxial displacement of the excited ceramic. I am asking about the microstructural effect.

Reviewer #3 (Remarks to the Author):

The manuscript has been improved. I am happy about it, except for one point: the question how much the electrostrictive response depends on an enhanced dielectric response. The article the authors cited for defending the value of 30 shows in fact that the $\epsilon_r < 30$ is in reality $\epsilon_r < 200$ to 500 or so. This should be corrected on line 45. I attached an annex with detailed argumentation and evidence from the cited ref. 16 (rebuttal letter). After this correction, the article can be published.

Reviewer #3 annex:

Question of dielectric response ϵ_{33} and electrostrictive M_{33} coefficient:

The authors decided not to change the statement with $\epsilon < 30$

44 factor of 10 below that of PMN-PT, the best electrostrictor currently in use^{9,10}. However, between
45 1-100 Hz, M_{33} for the doped ceria ceramics coexists with low relative dielectric permittivity $\epsilon < 30$
46¹¹ (vs $\epsilon_{PMN-PT} > 10,000$) and elastic modulus twice as large as that of PMN-PT. The combination

They write in their response letter:

For some compositions of aliovalent doped ceria, e.g. Lu- or Yb- doped ceria, the longitudinal electrostriction strain coefficient is $\approx 10^{-17} \text{ m}^2/\text{V}^2$, which is only approximately a factor of 10 below that of PMN-PT, the best electrostrictor currently in use^{35,36}. However, between 1-100 Hz, M_{33} for the doped ceria ceramics coexists with low relative dielectric permittivity $\epsilon < 30$ ¹⁶ (vs. $\epsilon_{PMN-PT} > 10,000$) and elastic modulus more than twice as large as that of PMN-PT.

In the response letter of the authors, they cite ref 16 (Kabir et al 2020) for showing that the dielectric constant does not increase like M_{33} with lower frequency. First, this paper does not show the frequency dependence of M_{33} , but only the one of the dielectric response ϵ_{33}' (real part, thus not conduction). M_{33} is measured at 1 Hz, the ϵ_{33}' given in table 1 are, however, measured at around 10 kHz. Figure 2 of this article reports the dielectric response. Accordingly, the column for ϵ_{33}' (1 Hz) should be 160/ 150/150/> 400 (see below). What is clear is that the fluorites apparently have larger microscopic electrostrictive Q constants than the ferroelectric perovskites, which are related to the M -constants like: $M=Q\epsilon^2$. Simple arguments based on ion displacements in electric field show that $Q \propto \epsilon^{-1}$ (Newnham papers). Conclusions can only be made at identical frequencies, particularly when the frequency dependence is large as in these materials (due to slow ion hopping).

FIGURE REDACTED

In citations 35 and 36 there are no measurements of the dielectric constant presented.

I still request that when citing dielectric constant values, it must be values measured at the same frequency as for the electrostrictive constants. Their statement on line 44 should consider reality and be changed to $\epsilon < 200 \dots 500$, which is still much less than the one of PMN-PT.

Color codes for fonts:

- black text = reviewer comments;
- green text = replies;
- yellow highlighting on green text = changes to the manuscript or SI.

Some changes were added to improve clarity:

- The title of the y-axis in Figure S4-2 was changed to M (A/m) from $M_{w/La} - M_{w/oLa}$ (A/m), because La doped samples were removed in previous revision.
- In previous version we did not clarify how Van Vleck paramagnetism parameter (χ_0) was obtained. We have added the clarification to the SI and experimental section.
- In Figure S5-2 some experimental details were added.
- In Figure S5-3 the x-axis label was misspelled and fixed in the revision.

Reviewer #1 (Remarks to the Author):

Reviewer #1, Comment 1:

The authors have addressed all the points mentioned in my report. I hope they helped improve the manuscript that can now be published.

Reviewer #1, Response 1:

We thank the reviewer for the decision in favor of publication.

Reviewer #2 (Remarks to the Author):

The authors have been responsive to some of my comments very well and improved some parts of the manuscript significantly. The main discussion on the generation of large electrostriction in Zr-doped Ceria (\leq Zr 12.5%) remains the same. I noticed that in the revised manuscript the authors attempted to make the results in a line by excluding certain data presented in the previous version of the manuscript, e.g., the conductivity (at $100 \text{ mHz} < f < \text{MHz}$) and permittivity (at $100 \text{ Hz} < f < \text{MHz}$) of the oxidized $\text{Zr}_{0.1}\text{Ce}_{0.9}\text{O}_2$ and reduced $\text{Zr}_{0.1}\text{Ce}_{0.9}\text{O}_{2-x}$, measured at $100 \text{ mHz} < f < \text{MHz}$ (in Fig. S5-2). The proposed mechanism behind the observed electrostriction has become much clearer now thanks to other reviewers' help. It is indeed an interesting finding that the large electrostrictive properties (effective $M_{33}^* \sim 10^{-16} \text{ m}^2/\text{V}^2$) can dominantly occur in cubic fluorite CeO_2 structure by incorporating isovalent dopants smaller than Ce^{4+} , without a significant contribution of oxygen vacancies. In particular, the large electrostriction of $\text{Zr}_{0.1}\text{Ce}_{0.9}\text{O}_2$ is extended up to a few kHz unlike large electrostriction in aliovalent cation doped CeO_{2-x} , usually limited by $f < 10 \text{ Hz}$. I agree now that this work may deliver a significant material development on fluorite-based electrostrictors scientifically and technologically if it is non-classical as the authors just claim. I should mention that the electromechanical properties of the just named non-classical/giant electrostrictors are not completely established yet. The electromechanical properties of non-classical/nonlinear electrostrictors are mostly based on the motion of charged point defects/ions. Thus, the situation is different with $\text{Zr}_{0.1}\text{Ce}_{0.9}\text{O}_2$ [stiff ZrO_8 (no increase in M at low frequency), the absence of mobile (/dynamical) defects, and no static elastic dipoles], compared to other non-classical electrostrictors (e.g., aliovalent cation-doped CeO_{2-x}). It is still an intriguing puzzle that $\text{Zr}_{0.1}\text{Ce}_{0.9}\text{O}_2$ also shows a small permittivity (experimental), but a large electrostriction.

To clarify this, I (also other two reviewers) asked the apparent permittivity of the $\text{Zr}_x\text{Ce}_{1-x}\text{O}_2$ samples under the same electric fields, which were applied to the electromechanical measurements of the samples (although the authors present the converse effect now). The authors could not measure the permittivity of the samples in the same field conditions (due to their technical issue), applied for their electromechanical measurements. Also, in the authors' reply, some of my last comments (e.g., the conductivity/imaginary part of complex dielectric constants in frequency as a function of Zr%) were just ignored. They are all important for readers!

In summary, the research finding reported here is interesting and important to the related research community. However, I recommend that the authors address the points below before considering publication.

Reviewer #2, Comment 1:

1. It is fine to provide a perspective on the impact of cation size mismatch in the discussion/summary section. However, the title of the manuscript is too general. Since the authors investigated only one material system, it should be specified with $Zr_xCe_{1-x}O_2$ in the title.

Reviewer #2, Response 1:

We agree with the reviewer's comment. We have modified the title to emphasize that we have found a material that displays certain properties. Therefore, the title has been changed to:

“Lead-free, low permittivity Zr-doped ceria ceramics displaying giant electrostriction”

Reviewer #2, Comment 2:

2.1. In my view, the electrostrictive strain doesn't linearly increase with E^2 in Fig. 2a although the newly updated Fig. S6-2 shows linearity with the converse effect. The authors used a large symbol size for plotting the data in Fig. 2a. Please confirm this with error bars. If not perfectly linear, the word, "linear" should be changed as to "almost linear". That is why I previously asked about the appearance of non-linearity with additional effect (e.g., oxygen vacancies, also see a paramagnetic response in the oxidized Zr10% sample).

2.2. Can the authors completely rule out the oxygen vacancy effect from their experiments to deliver a concrete conclusion?

2.3. Also, if the Zr doped ceria samples have an electrically nonlinear nature, what can we expect from?

Reviewer #2, Response 2:

The reviewer's question has three parts, and we present our answers accordingly.

2.1. The figure has been changed in order to improve the visibility of the error bars, and the goodness of fit parameter (R_{adj}^2) has now been added to the caption. The new Figure 2a and caption are given below.

Figure 2. (a) Longitudinal electrostrictive strain measured for $Zr_{0.1}Ce_{0.9}O_2$ as a function of the applied electric field squared. Values of R_{adj}^2 for the linear fit for strain measurements made at 0.15Hz, 10Hz, 100Hz are 0.97, 0.93, 0.95, respectively. Measurements were made in triplicate for each sample (>4 samples for each composition) under ambient conditions; in some cases, error bars are smaller than the symbols. All samples contract parallel to the applied field.

A change was made in text:

The dependence of strain on electric field, u_{33} vs E_3^2 remains near-linear for $E \leq 13.4$ kV/cm, where $u_{33} \approx -200$ ppm (Error! Reference source not found.a)

2.2. The high sensitivity of SQUID measurements allows one to reliably estimate the concentration of Ce^{3+} , and by extension, also the concentration of oxygen vacancies, in oxidized samples with <10mol% Zr as being < 100 ppm. At such a low level, their effect on electrostriction should be negligible. The general question of the role played by oxygen vacancies in non-classical electrostrictors, including both aliovalent and isovalent -doped ceria, is currently under study, the results of which will be reported elsewhere.

2.3. For the material presented in the manuscript, the electrostrictive strain is near-linear with the field squared (see 2.1 above). The local electric field (electric field felt by the elastic dipoles) is proportional to the external field with a factor $(\epsilon' + 2)/3$ (if the Lorenz factor is $1/3$). Therefore, the linearity of the electrostrictive strain with field squared, implies that ϵ' does not depend on the field. Therefore, we did not investigate non-linearity.

Reviewer #2, Comment 3:

3. The authors addressed it with a limit of dipolar switching (dielectric relaxation) of ZrO₈ at high frequencies (a few kHz). After that, the conductivity (ϵ'') of the samples increases. I can understand that it is challenging to assign the conducting source in the samples with Zr% (e.g., an increased structural/lattice distortion, defect formation and electronic contribution). However, to emphasize and manifest the Zr doping effect in ceria, the authors should present the frequency dependent M_{33} of Zr-doped ceria samples as a function of Zr% in the frequency range of 100 mHz (or 10 Hz) $< f < 1$ MHz and incorporate them in Fig. 2b. I am guessing the damping (/relaxation) frequency of M increases up to Zr = 12.5 % and then it will decrease after 12.5 %. This variation can be compared with the frequency-dependent conductivity variation with Zr%.

Reviewer #2, Response 3:

Figure 2b has been modified in order to include the direct and converse measurements for undoped and 5mol% Zr doped ceria:

Figure 2. (b) Log-log plot of the absolute value of the direct and the converse ($V_{AC} = 10$ V) longitudinal electrostriction strain coefficients for undoped ceria and for 5-10 mol% Zr doped ceria as a function of frequency.

Samples with >10 mol% Zr were found to have a high concentration of oxygen vacancies, even following oxidation. Therefore, measurements of these samples are not included in Fig. 2b.

Reviewer #2, Comment 4:

In Fig. S9 of the revised manuscript, there is a phase lag for the 2nd harmonic displacement response of a $\text{Zr}_{0.1}\text{Ce}_{0.9}\text{O}_2$ sample in response to the applied AC voltage in time. What is its origin? Does the phase lag vary with frequency? Also, how long does the mechanical response of the electrically excited sample last (endurance)? I strongly suggest including long-time measurement data, either in the main text or in SI, to further emphasize the large and sustainable electrostriction of the developed ceria. Also, plot a strain Vs applied voltage curve and describe the origin of the phase lag in the figure caption.

Reviewer #2, Response 4:

The phase lag between the 2nd harmonic displacement in response to the applied AC voltage, as drawn in Fig. S9, is not an actual, physical phase lag. The phase lag is introduced between the amplifier voltage monitor port, and the output of the capacitance meter. Both ports are connected to a Keithley 2000 digital multimeter from which data are captured by a VISA interface. The apparent phase lag is a result of the phase shift introduced by the presence of the multimeter, device interrogation sequence, and delay between the interrogation instances and the measurements. The function of the voltage monitor port is to provide the shape and amplitude of the applied voltage. This point is now clarified in the revised figure caption.

Figure S9. Real time strain and voltage measurements during an electrostriction experiment at $f = 0.15$ Hz, $V = 1550$ V, 10 mol% Zr-doped (oxidized) ceria pellet. The phase lag does not represent an actual delay between the strain and applied voltage. The phase lag is introduced between the amplifier voltage monitor port, and the output of the capacitance meter. Both ports are connected to a Keithley 2000 digital multimeter from which data are captured by a VISA interface. The apparent phase lag is a result of the phase shift introduced by the presence of the multimeter, device interrogation sequence, and delay between the interrogation sequence and the measurements.

With respect to sample endurance, the amplitude of the response does not change even following >72hrs of continuous application of the maximum electric field (13.4 kV/cm) used in this study.

Reviewer #2, Comment 5:

5. In Fig. S10 of the revised manuscript, the M_{33} of the reduced sample in the frequency range of 100 mHz – 100 Hz shows much lower than that of the oxidized one. But, in Fig. S10 of the first version of the manuscript, the conductivity of the reduced and oxidized samples is very similar in the measured frequency range. Also, in the first version, the dielectric permittivity of the reduced sample at $f = 100$ Hz is larger than that of the oxidized one. The data are not consistent! Why don't they agree with each other? This is very strange to me.

Reviewer #2, Response 5:

This is figure S10 in the revised manuscript -R1, with caption:

Figure S10. Frequency dependence of the direct, longitudinal electrostriction coefficient $|M_{33}|$ of sintered ceria pellets doped with Zr^{4+} , both before (Red) and after (Ox) re-oxidation; Measurements were made in triplicate under ambient conditions; in some cases, error bars are smaller than the symbols.

This is Fig.S5-2 in the original manuscript:

This is Fig. S5-2 in the revised- R1 manuscript with caption:

Figure SError! No text of specified style in document.-1. Room temperature (a) conductivity and (b) real component of the relative dielectric permittivity for Zr-doped ceria pellets 0 V_{DC} bias, 10 V_{AC},

frequency range 1 MHz-100Hz, measured with spring loaded electrodes. Measurements were made in the same device as the converse electrostriction measurements.

The data for the non-oxidized samples were removed from the figures in keeping with our response to reviewer #2, comment #9 concerning the original submission, that the poorly controlled oxygen vacancy concentration in non-oxidized samples does not allow reliable electrical and electrostriction measurements.

Indeed, the dielectric permittivity ϵ' measured for the non-oxidized samples is higher than that measured for the oxidized samples (S5-2 original manuscript) despite the fact that the electrostriction strain coefficient of the former is lower. As was noted in the original submission, electrostriction measurements on reduced samples are not reliable: reduction introduces a number of species (Ce^{3+} , oxygen vacancies) along with polaronic electron hopping, each of which may affect the electrostrictive response differently. Therefore, upon the advice of reviewer #2 (comment #9), the reduced samples were removed from further consideration. Understanding the effect of reduction on electromechanical response is a topic well beyond the scope of the current work.

Reviewer #2, Comment 6:

6. The SQUID data shows that there is still a paramagnetic-like response even in the oxidized sample. What is the origin of this paramagnetism in the sample? Is it because of the presence of oxygen vacancies in an irregular manner? If so, both the oxidized and reduced samples possess oxygen vacancies with different amounts. The authors just mention a known defect concentration in SI. The valence state of Ce ions can also be checked by EELS and XAS (Ce L-edge spectra, for example, Wu et al., Physica Scripta. T115, 802 (2005)). Have the authors measured the XAS Ce L-edge spectra of the undoped and doped ceria samples?

Reviewer #2, Response 6:

Paramagnetism in ceria may be due to: a small concentration of Ce^{3+} ; residual magnetic impurities such as iron; as well as Van Vleck temperature-independent paramagnetism. Even when the presence of Ce^{3+} is negligible, for instance, suppressed by aliovalent doping, ceria can nevertheless display Van Vleck paramagnetic response as we have shown previously [1, 2] This response does not interfere with concentration measurements of Ce^{3+} and/or oxygen vacancies.

XAS of Zr doped ceria ceramics measured at the Ce L_3 -edge and reported in Fig. 4a, is consistent with the majority of Ce atoms being in the Ce^{4+} oxidation state. The revised figure 4a (to which we have now added a reference spectrum of undoped ceria, in accordance with the reviewer's comment) shows that the pre-edge and rising edge of Ce L_3 XANES data in doped ceria are well aligned with that of undoped ceria, ruling out (at least by XANES) any detectable amount of Ce^{3+} .

We have now added the XANES spectrum for undoped ceria to Figure 4a .

The highlighted sentence has now been added to the manuscript immediately preceding the sentence: “The local environment of the Zr atoms is analyzed by combining the Zr K-edge XANES and the extended X-ray absorption fine structure (EXAFS)”:

The putative presence of Ce^{3+} at the rising edge of the Ce XANES spectrum could not be detected for any of the samples.

The figure caption now reads:

Figure 1. Normalized Ce L_3 edge XANES spectra (a) and Zr K-edge EXAFS spectra in k-space (b) and r-space (c) for 5, 10 and 20 mol% Zr doped ceria powders. r-space spectra were obtained by Fourier transforming the k^2 -weighted $\chi(k)$ spectra in the k range 3-14.5 \AA^{-1} . The XANES spectrum of

undoped ceria powder is included as a reference in (a). Details of XAS measurements are presented in the Online Methods section.

Reviewer #2, Comment 7: 7.

Lastly, all the ceramic samples studied here are polycrystalline, as shown in XRD. This means that grains oriented along the (111), (200), (220), and (311) crystallographic directions can be excited under uniaxial electric field applications. The authors claim that the motion of ZrO₈ is the key to generating large electromechanical displacement (negative longitudinal electrostrictive strain) in Zr doped ceria. In Fig. 6d, there are positive and negative displacements in the ZrO₈ with different crystallographic directions with a positive Poisson's ratio. In the experiment, all grain orientations cannot be aligned by electric field. I wonder how Zr/O vibrations in differently oriented grains respond to electric field application and how they generate the corresponding uniaxial displacement of the excited ceramic. I am asking about the microstructural effect.

Reviewer #2, Response 7:

Indeed, the fact that we only measured M_{33} in a polycrystalline sample must influence the magnitude of the experimentally measured displacement. Modelling (Fig. 6a) predicts that force-displacement curves for Zr-O displacement are almost the same for [100], [110] and [111] directions (varying from 2.47 eV/Å² along the [111] direction to 3.01 eV/Å² along the [100] direction as shown in Table S8-2). Thus, Zr displacement along different crystallographic directions is almost isotropic. Only the elastic dipole tensor induced by the Zr-[111]-direction displacement was calculated in Fig 6d, as we expect M_{33} to be almost isotropic as well. We agree with the reviewer that “all grain orientations cannot be aligned by electric field”, therefore the predicted electrostriction strain must be considered to be an upper bound, as we stated in the manuscript: “This value is consistent with our estimate of an upper bound on Zr displacement” and “we suggest that not all elastic dipoles are aligned under the field.” Detailed study of the dependence of M_{ij} on crystallographic direction and microstructure is beyond the scope of this work.

Reviewer #3 (Remarks to the Author):

The manuscript has been improved. I am happy about it, except for one point: the question how much the electrostrictive response depends on an enhanced dielectric response. The article the authors cited for defending the value of 30 shows in fact that the $\epsilon < 30$ is in reality $\epsilon < 200$ to 500 or so. This should be corrected on line 45. I attached an annex with detailed argumentation and evidence from the cited ref. 16 (rebuttal letter). After this correction, the article can be published.

Question of dielectric response ϵ_{33} and electrostrictive M_{33} coefficient:

The authors decided not to change the statement with $\epsilon < 30$

- 44 factor of 10 below that of PMN-PT, the best electrostrictor currently in use^{9,10}. However, between
45 1-100 Hz, M_{33} for the doped ceria ceramics coexists with low relative dielectric permittivity $\epsilon < 30$
46¹¹ (vs $\epsilon_{PMN-PT} > 10,000$) and elastic modulus twice as large as that of PMN-PT. The combination

They write in their response letter:

For some compositions of aliovalent doped ceria, e.g. Lu- or Yb- doped ceria, the longitudinal electrostriction strain coefficient is $\approx 10^{-17} \text{ m}^2/\text{V}^2$, which is only approximately a factor of 10 below that of PMN-PT, the best electrostrictor currently in use^{35,36}. However, between 1-100 Hz, M_{33} for the doped ceria ceramics coexists with low relative dielectric permittivity $\epsilon < 30$ ¹⁶ (vs. $\epsilon_{PMN-PT} > 10,000$) and elastic modulus more than twice as large as that of PMN-PT.

In the response letter of the authors, they cite ref 16 (Kabir et al 2020) for showing that the dielectric constant does not increase like M_{33} with lower frequency. First, this paper does not show the frequency dependence of M_{33} , but only the one of the dielectric response ϵ_{33}' (real part, thus not conduction). M_{33} is measured at 1 Hz, the ϵ_{33}' given in table 1 are, however, measured at around 10 kHz. Figure 2 of this article reports the dielectric response. Accordingly, the column for ϵ_{33}' (1 Hz) should be 160/150/150/> 400 (see below). What is clear is that the fluorites apparently have larger microscopic electrostrictive Q constants than the ferroelectric perovskites, which are related to the M -constants like: $M=Q\epsilon^2$. Simple arguments based on ion displacements in electric field show that $Q \propto \epsilon^{-1}$ (Newnham papers). Conclusions can only be made at identical frequencies, particularly when the frequency dependence is large as in these materials (due to slow ion hopping).

[FIGURE REDACTED]

In citations 35 and 36 there are no measurements of the dielectric constant presented.

I still request that when citing dielectric constant values, it must be values measured at the same frequency as for the electrostrictive constants. Their statement on line 44 should consider reality and be changed to $\epsilon < 200 \dots 500$, which is still much less than the one of PMN-PT.

Reviewer #3, Response 1:

We appreciate the reviewer's note and certainly agree that the “effective” dielectric permittivity of ionic conductors increases significantly with frequency due to ionic transport-induced polarization. The work cited by the reviewer (Kabir *et al.*) in fact provides the frequency dependence of both the relative dielectric permittivity and the longitudinal electrostriction strain coefficient for aliovalent doped ceria.

[FIGURE REDACTED]

In response to the reviewer’s comment, we have modified the introductory text:

For aliovalent doped ceria (CeO_2), an electrostriction strain coefficient as large as that of PMN-PT ($10^{-16} \text{ m}^2/\text{V}^2$) [3, 4], coexists with a relative dielectric permittivity that is much lower than that of PMN-PT, as well as with a much higher elastic modulus. However, only at very low frequencies ($< 1 \text{ Hz}$), where the relative dielectric permittivity is $30 < \epsilon' < 500$ is the longitudinal electrostriction strain coefficient of doped ceria comparable to that of PMN-PT. At higher frequencies, both the electrostriction strain coefficient and the dielectric permittivity decrease to $10^{-17} - 10^{-18} \text{ m}^2/\text{V}^2$ and < 50 , respectively. In both frequency ranges, this combination of relatively low permittivity and high elastic modulus (*i.e.*, weak compliance) places $|Q_h|$ calculated for doped ceria at least two orders of magnitude above that predicted by Newnham’s scaling law [5, 6] and identifies it, as well as other recently described ceramics with a large concentration of point defects [6-11], as “giant”, or non-classical, electrostrictors [5, 12].

Also please note the frequency dependence of 10mol% Yb-doped ceria: (a) dielectric permittivity ϵ' and electrostriction strain coefficients, and (b) measured and calculated hydrostatic electrostriction polarization coefficients (Young’s modulus 210 GPa, Poisson ratio 0.31).

References

1. Varenik, M., et al., *Van Vleck paramagnetism in undoped and Lu-doped bulk ceria*. Physical Chemistry Chemical Physics, 2018. **20**(42): p. 27019-27024.
2. Ackland, K. and J.M.D. Coey, *Room temperature magnetism in CeO₂-A review*. Physics Reports-Review Section of Physics Letters, 2018. **746**: p. 1-39.
3. Kabir, A., et al., *Enhanced Electromechanical Response in Sm and Nd Co-doped Ceria*. Materialia, 2020. **12**: p. 100728.
4. Varenik, M., et al., *Dopant Concentration Controls Quasi-Static Electrostrictive Strain Response of Ceria Ceramics*. ACS applied materials & interfaces, 2020. **12**(35): p. 39381-39387.
5. Newnham, R., et al., *Electrostriction: nonlinear electromechanical coupling in solid dielectrics*. The Journal of Physical Chemistry B, 1997. **101**(48): p. 10141-10150.
6. Yavo, N., et al., *Large Nonclassical Electrostriction in (Y, Nb)-Stabilized delta-Bi₂O₃*. Advanced Functional Materials, 2016. **26**(7): p. 1138-1142.
7. Santucci, S., et al., *Electro-chemo-mechanical effect in Gd-doped ceria thin films with a controlled orientation*. Journal of Materials Chemistry A, 2020. **8**(28): p. 14023-14030.
8. Santucci, S., et al., *Electromechanically active pair dynamics in a Gd-doped ceria single crystal*. Physical Chemistry Chemical Physics, 2021. **23**(19): p. 11233-11239.
9. Tinti, V.B., et al., *Gigantic electro-chemo-mechanical properties of nanostructured praseodymium doped ceria*. Nanoscale, 2021. **13**(16): p. 7583-7589.
10. Han, J., et al., *Enhanced electromechanical properties in low-temperature gadolinium-doped ceria composites with low-dimensional carbon allotropes*. Journal of Materials Chemistry A, 2022. **10**(8): p. 4024-4031.
11. Makagon, E., et al., *Non-Classical Electrostriction in Hydrated Acceptor Doped BaZrO₃: Proton Trapping and Dopant Size Effect*. Advanced Functional Materials, 2021. **31**(50): p. Artn 2104188.
12. Yu, J.C. and P.E. Janolin, *Defining "giant" electrostriction*. Journal of Applied Physics, 2022. **131**(17).

REVIEWERS' COMMENTS

Reviewer #2 (Remarks to the Author):

The authors have mostly responded to my comments although some aspects remain unclear yet (e.g., the imaginary part of complex dielectric permittivity with apparent field applications, phase lag, and long-term measure). The authors have assigned these remaining parts to future work. Overall, I am happy with the new inputs and corrections that have improved the manuscript. The manuscript delivers significance in the research fields, so I recommend it for publication in Nat. Commun.

Reviewer #3 (Remarks to the Author):

Rev.2: "The proposed mechanism behind the observed electrostriction has become much clearer now thanks to other reviewers' help. It is indeed an interesting finding that the large electrostrictive properties (effective $M_{33}^* \sim 10^{-16} \text{ m}^2/\text{V}^2$) can dominantly occur in cubic fluorite CeO_2 structure by incorporating isovalent dopants smaller than Ce^{4+} , without a significant contribution of oxygen vacancies....I agree now that this work may deliver a significant material development on fluorite-based electrostrictors scientifically and technologically if it is non-classical as the authors just claim."

On this critical remarks is "The authors could not measure the permittivity of the samples in the same field conditions (due to their technical issue), applied for their electromechanical measurements."

This is indeed the case. For the dielectric measurements the AC voltage is defined: $V_{ac} = 10 \text{ V}$. In fig. S9, we see the voltage applied for the strain measurement: $V_{ac} = 1550 \text{ V}$. This is 2 orders of magnitude higher. The good point of the work is that they compared with the electrostrictive response of PMN-PT and with the piezoelectric response of quartz and obtained correct values. This means that the results obtained at high V_{ac} is ok. The M_{33} they measured is correct. The question is how much the M_{33} depends on the V_{ac} voltage. Their problem is that they measure at bulk ceramics, and they need much higher voltages than with thin films. The commercial impedance meters for testing dielectrics are very limited in voltage, and the used 10 V is probably already the maximum. One of their

statement is that the dielectric constant remains low. I think this statement is correct. A lower AC voltage leads rather to higher dielectric constants because Schottky barriers at the metal interface, and a varactor effect (insulating grain boundaries) could be present, introducing high capacitances at the interfaces. So I think that increasing the AC voltage in the dielectric measurement rather leads to lower values of the real part. Hence, their claim is correct. However, I agree that the AC voltage for strain measurements should be defined in the text, either in the figure caption, or the description of the experimental setup. On this way the experiments can be reproduced by others.

After having added the V_{ac} for strain measurements, the paper should be accepted, in my opinion.

Color codes for fonts:

- black text = reviewer comments;
- green text = replies;
- yellow highlighting on green text = changes to the manuscript or SI.

Other changes:

1. Grant number was corrected:

The work on development of novel electrostrictive materials was supported by the US Army Research Office (ARO grant #W911NF2110263, I.L.). The work on development of the theoretical description and the properties of the elastic dipoles was supported by the Israel-US Binational Science foundation regular program (Y.Q. + I.L grant # 2020108). IL and AIF acknowledge the NSF-BSF program grant 2022786, which supported the synchrotron measurements and their analysis. AIF and PKR acknowledge support by NSF Grant number DMR-2312690. This research used beamline 7-BM (QAS) of the National Synchrotron Light Source II (NSLS-II), a U.S. DOE Office of Science User Facility operated for the DOE Office of Science by Brookhaven National Laboratory under contract no. DE-SC0012704. We gratefully acknowledge Drs. Lu Ma and Steven Ehrlich for their support of the XAS measurements at the 7-BM beamline. We acknowledge support of the beamline experiments by the Synchrotron Catalysis Consortium funded by the US Department of Energy, Office of Science, Office of Basic Energy Sciences, Grant No. DE-SC0012335. AIF acknowledges support by Weston Visiting Professorship during his stay at the Weizmann Institute of Science.

2. Other changes are listed in the author guidelines.

Reviewer #2 (Remarks to the Author):

The authors have mostly responded to my comments although some aspects remain unclear yet (e.g., the imaginary part of complex dielectric permittivity with apparent field applications, phase lag, and long-term measure). The authors have assigned these remaining parts to future work. Overall, I am happy with the new inputs and corrections that have improved the manuscript. The manuscript delivers significance in the research fields, so I recommend it for publication in Nat. Commun.

Reviewer #2, Comment #1:

We thank the reviewer for their decision.

Reviewer #3 (Remarks to the Author):

Rev.2: "The proposed mechanism behind the observed electrostriction has become much clearer now thanks to other reviewers' help. It is indeed an interesting finding that the large electrostrictive properties (effective $M_{33}^* \sim 10^{-16} \text{ m}^2/\text{V}^2$) can dominantly occur in cubic fluorite CeO_2 structure by incorporating isovalent dopants smaller than Ce^{4+} , without a significant contribution of oxygen vacancies....I agree now that this work may deliver a significant material development on fluorite-based electrostrictors scientifically and technologically if it is non-classical as the authors just claim."

Reviewer #3, Comment #1:

We thank the reviewer for their decision.

On this critical remarks is "The authors could not measure the permittivity of the samples in the same field conditions (due to their technical issue), applied for their electromechanical measurements."

This is indeed the case. For the dielectric measurements the AC voltage is defined: $V_{ac} = 10 \text{ V}$. In fig. S9, we see the voltage applied for the strain measurement: $V_{ac} = 1550 \text{ V}$. This is 2 orders of magnitude higher. The good point of the work is that they compared with the electrostrictive response of PMN-PT and with the piezoelectric response of quartz and obtained correct values. This means that the results obtained at high V_{ac} is ok. The M_{33} they measured is correct. The question is how much the M_{33} depends on the V_{ac} voltage. Their problem is that they measure at bulk ceramics, and they need much higher voltages than with thin films. The commercial impedance meters for testing dielectrics are very limited in voltage, and the used 10 V is probably already the maximum. One of their statement is that the dielectric constant remains low. I think this statement is correct. A lower AC voltage leads rather to higher dielectric constants because Schottky barriers at the metal interface, and a varactor effect (insulating grain boundaries) could be present, introducing high capacitances at the interfaces. So I think that increasing the AC voltage in the dielectric measurement rather leads to lower values of the real part. Hence, their claim is correct. However, I agree that the AC voltage for strain measurements should be defined in the text, either in the figure caption, or the description of the experimental setup. On this way the experiments can be reproduced by others.

After having added the V_{ac} for strain measurements, the paper should be accepted, in my opinion.

Reviewer #3, Comment #2:

We thank the reviewer for their decision.

The V_{AC} values for strain measurements were added in:

1. The caption to figure 2:

Figure 1. Electrostrictive response measured for oxidized ceria ceramics as described in the Online Methods section: **(a)** Longitudinal electrostrictive strain measured for $Zr_{0.1}Ce_{0.9}O_2$ as a function of the applied electric field squared. Values of R^2_{adj} for the linear fit for strain measurements made at 0.15Hz, 10Hz, 100Hz are 0.97, 0.93, 0.95, respectively. Typical amplitude for the AC voltage applied on the samples were between $100 V_{AC}$ to $1750 V_{AC}$ **(b)** Log-log plot of the absolute value of the converse ($V_{AC} = 10 V$) and direct longitudinal electrostriction strain coefficients for undoped ceria and for 5 - 10mol% Zr doped ceria as a function of frequency. **(c)** Direct longitudinal electrostriction strain coefficient of Zr-doped ceria as a function of Zr content, $f \geq 100Hz$. Measurements were made in triplicate for each sample (>4 samples for each composition) under ambient conditions; in some cases, error bars are smaller than the symbols. All samples contract parallel to the applied field.

2. A sentence was added to the methods section under **Measurements of the direct electrostriction effect: field-induced strain:**

Briefly, the ceramic pellet was inserted between two stainless steel electrodes, with the top electrode being spring loaded. The displacement, δ , of the top surface under voltage was measured with a proximity sensor (± 0.02 nm) monitored with a lock-in amplifier. Alternating sine-wave voltage was applied to the sample and only second harmonic response was detected. The electrostriction coefficient in such a configuration is given by: $M_{33} = \delta \cdot th / V^2$, where V is the voltage applied and th is the sample thickness measured with accuracy $\pm 2 \mu m$. The values of strain and electric field were calculated as $u = \frac{\delta}{th}$ and $E = V/th$. The measurements were performed under ambient conditions (297 ± 2 K, relative humidity 20% - 55%). Typical amplitude for the AC voltage applied on the samples were between $100 V_{AC}$ to $1750 V_{AC}$, and typical sample thickness as 1 mm.